# CCl$_4$ emissions in eastern China during 2021–2022 and exploration of potential new sources

Bowei Li[1], Jiahuan Huang[2], Xiaoyi Hu[1], Lulu Zhang[2], Mengyue Ma[1], Liting Hu[1], Di Chen[1], Qianna Du[1], Yahui Sun[1], Zhouxiang Cai[1], Ao Chen[3], Xinhe Li[1], Rui Feng[1], Ronald G. Prinn[4] & Xuekun Fang[1,4] ✉

According to the Montreal Protocol, the production and consumption of ozone-layer-depleting CCl$_4$ for dispersive applications was globally phased out by 2010, including China. However, continued CCl$_4$ emissions were disclosed, with the latest CCl$_4$ emissions unknown in eastern China. In the current study, based on the atmospheric measurements of ~12,000 air samples taken at two sites in eastern China, the 2021–2022 CCl$_4$ emissions are quantified as $7.6 \pm 1.7$ gigagrams per year. This finding indicates that CCl$_4$ emissions continued after being phased out for dispersive uses in 2010. Subsequently, our study identifies potential industrial sources (manufacture of general purpose machinery and manufacture of raw chemical materials, and chemical products) of CCl$_4$ emissions.

Carbon tetrachloride (CCl$_4$) is a first-generation ozone-depleting substance (ODS) with an ozone depletion potential of 0.87 and a potent greenhouse gas with a global warming potential of 2150[1]. According to the Montreal Protocol and its amendments, since 1995, the production and consumption of CCl$_4$ for dispersive uses (i.e., solvents, cleaning agents, and process agents) should have been phased out gradually with 100% reduction by 2010[2,3]. An exemption was its use as a chemical feedstock or processing agent[4]. Top-down estimates of global emissions, inferred from atmospheric observations, have shown no significant decrease in emissions of CCl$_4$ between 2010 and 2019 (decline rate of $0.1 \pm 0.2$ Gg yr$^{-1}$)[5]. However, a wide gap remains in the emission estimates of CCl$_4$. For instance, the bottom-up emission inventory estimates (obtained by multiplying the activity data and emission factors) imply approximately 26.1 (12.6–40.0) Gg yr$^{-1}$ emissions in 2019, 17 Gg yr$^{-1}$ lower than those of the top-down emission estimates (43 Gg yr$^{-1}$ in 2019)[1]. Therefore, further atmospheric observations of CCl$_4$ are needed to understand the reason for the gap in bottom-up and top-down emissions.

In 1991, China signed the Montreal Protocol, and in 1993[6] issued a plan for the production and consumption phase-out of ODSs.

Moreover, the application of CCl$_4$ for dispersive uses was to be phased out by 2010. A bottom-up emission inventory showed that in China, emissions of CCl$_4$ from dispersive uses were reduced from 9.3 Gg yr$^{-1}$ in 2005 to 1.1 Gg yr$^{-1}$ in 2010 and to zero afterwards[7]. Meanwhile, a bottom-up inventory for non-emissive uses of CCl$_4$, including coal combustion, tetrachloroethylene conversion, feedstock, laboratory chemical reagents, and CCl$_4$ by-product production, showed that the emissions were reduced from 13 Gg yr$^{-1}$ in 2005 to 4 Gg yr$^{-1}$ in 2011, and remained unchanged between 2011 and 2014 (4–5 Gg yr$^{-1}$)[8]. Moreover, another independent study showed bottom-up emission estimates for CCl$_4$ in 2014 of 7.3 Gg yr$^{-1}$ after including non-feedstock emissions from chloromethane and perchloroethylene plants[9]. Nevertheless, top-down estimates of CCl$_4$ emissions based on field atmospheric observations indicated substantial emissions in China after 2010 ($8.8 \pm 6.4$ Gg yr$^{-1}$ during 2010–2011[10], $23.6 \pm 7.1$ Gg yr$^{-1}$ during 2011–2015[11], and 13 (7–19) Gg yr$^{-1}$ during 2009–2016[12]) that were substantially higher than the bottom-up emission estimates. Thus, wide gaps exist in top-down and bottom-up emission estimates of CCl$_4$; however, the missing sources remain unclear.

[1]College of Environmental & Resource Sciences, Zhejiang University, 310058 Hangzhou, Zhejiang, China. [2]Wuxi Ecology Environment Monitoring and Control Center, 214062 Wuxi, Jiangsu, China. [3]Department of Environmental Health and Engineering, Johns Hopkins University, Baltimore, MD 21218, USA. [4]Center for Global Change Science, Massachusetts Institute of Technology, Cambridge, MA 02139, USA. ✉e-mail: fangxuekun@zju.edu.cn

Eastern China was identified as a hotspot of $CCl_4$ emissions[11–13], with an estimated emission of $10.9 \pm 2.0$ Gg yr$^{-1}$ from 2014 to 2017, which decreased to $6.2 \pm 1.4$ Gg yr$^{-1}$ in 2018–2019[14]. At least 40–60% of the unexpected increase of trichlorofluoromethane (CFC-11) global emissions during 2014–2017 was also attributed to eastern China with the likely production of CFC-11 and related $CCl_4$[15]. Based on the strong correlation between CFC-11 and $CCl_4$ ($R^2 = 0.78$), Benish et al.[13] showed the likely co-production of these two species. To strengthen the regulation of CFC-11 and the production of $CCl_4$, the Ministry of Ecology and Environment of China has dispatched working groups to perform on-site supervision and installed online monitoring systems for enterprises producing $CCl_4$ as a by-product in the country[16]. The "Supervision program" was implemented to improve the tracking of produced $CCl_4$, thereby preventing its conversion to CFCs (e.g., CFC-11) or its sale for controlled or emissive applications, which may lead to a decline in $CCl_4$ emissions from this source. $CCl_4$ is also used as feedstock, e.g., to produce pentafluoropropane (HFC-245fa) and fluoroolefins (HFOs) (these processes are not controlled by the Montreal Protocol)[17]. Since the feedstock use of $CCl_4$ for the production of HFCs, HFOs, and other chemicals has increased by ~70% between 2015 and 2019 in China[17], an increase in $CCl_4$ emissions from feedstock uses is expected. Given the combined effects of the above activities, the latest emissions of $CCl_4$ in eastern China are unclear. However, no data are available after 2019 on atmospheric concentrations and emissions of $CCl_4$ in eastern China. Furthermore, most atmospheric observation data used to estimate emissions of $CCl_4$ in previous top-down studies[11,12,14] were collected from the Gosan station (hereafter named GSN, 33.3° N, 126.2° E) in South Korea. The GSN station is more sensitive to northeastern China than middle and southeastern China (see discussion below). Thus, the emission information regarding $CCl_4$ from middle and southeastern China may not be adequately captured. Although there are reported atmospheric observations of $CCl_4$ within eastern China, namely within Nanjing ($120 \pm 30$ ppt; parts per trillion) in 2018[18], Dongying ($129 \pm 62$ ppt) in 2017[19], Hebei ($89 \pm 21$ ppt) in 2016[13], and Lin'an ($111 \pm 11$ ppt) in 2001[20], emission estimates of $CCl_4$ using these observations have not been reported. Therefore, an urgent need exists to determine the latest atmospheric concentrations and emissions of $CCl_4$ in eastern China.

Consequently, this study describes observations of atmospheric $CCl_4$ concentrations at two sites in eastern China (as discussed below, their spatial coverages are larger than the previously used GSN site) during 2021–2022. This study reports substantial emissions of $CCl_4$ in eastern China after the dispersive use of $CCl_4$ was phased out in 2010, and the government intensified $CCl_4$ regulations from 2019 to 2022. Finally, through an extensive industrial sampling campaign, we identified potential sources of emissions for $CCl_4$.

## Results and discussion
### $CCl_4$ emissions in eastern China
Two sites were set up to measure $CCl_4$ and other trace gases (Fig. 1). One observation site was located on the Zijingang Campus of Zhejiang University (hereafter named ZJU, 30.306°N, 120.095°E) in the northwest of Hangzhou City, Zhejiang Province in central-eastern China. The other sampling site was in the Damaojian Mountain in Shanghuang village (hereafter named SHH, 28.583°N, 119.508°E), Wuyi County, Zhejiang Province. Based on the emission sensitivity derived from the backward simulation using the FLEXible PARTicle dispersion (FLEXPART) model, ZJU and SHH were more sensitive to emissions from eastern China (especially in Zhejiang, Anhui, Jiangsu, Jiangxi, and Fujian provinces) than GSN (see Fig. 1 and Supplementary Fig. 2).

During the observation period, the average concentrations of $CCl_4$ were $79.3 \pm 14.3$ ppt and $80.0 \pm 14.6$ ppt at ZJU and $76.8 \pm 14.3$ ppt and $77.9 \pm 14.6$ ppt at SHH in 2021 and 2022, respectively (Fig. 2a and Supplementary Fig. 3). Using the atmospheric concentration observations at ZJU and SHH and an inverse modeling method, emissions of $CCl_4$ in eastern China were quantified (see Methods for detailed information). Eastern China in this study includes Shandong, Henan, Shanghai, Jiangsu, Zhejiang, Anhui, Jiangxi, and Fujian, whose emissions the mole fractions observed at ZJU and SHH sites were sensitive to (Fig. 1). Emissions of $CCl_4$ in eastern China were estimated to be $7.0 \pm 1.6$ Gg yr$^{-1}$ in 2021, and $8.2 \pm 1.8$ Gg yr$^{-1}$ in 2022, with a mean emission of $7.6 \pm 1.7$ Gg yr$^{-1}$ (Fig. 2b).

Park et al.[14] reported that emissions of $CCl_4$ in eastern China (see definition of boundary in Park et al.[14]) decreased from $11.9 \pm 1.5$ Gg yr$^{-1}$ in 2014 to $8.8 \pm 1.7$ Gg yr$^{-1}$ in 2017, $6.0 \pm 1.5$ Gg yr$^{-1}$ in 2018, and $6.3 \pm 1.0$ Gg yr$^{-1}$ in 2019 (Fig. 2b). Meanwhile, the emissions of $CCl_4$ in eastern China ($7.6 \pm 1.7$ Gg yr$^{-1}$) estimated in this study for 2021–2022 were comparable to (higher than, although not significantly) those in 2019 ($6.3 \pm 1.0$ Gg yr$^{-1}$)[14] within the uncertainties, suggesting persistent emissions of $CCl_4$ in these areas since 2019.

Moreover, emissions of $CCl_4$ in eastern China ($7.6 \pm 1.7$ Gg yr$^{-1}$) during 2021–2022 were significantly higher than the consumption of $CCl_4$ for dispersive uses (0.12 Gg yr$^{-1}$ in 2021) as reported by China to the United Nations Environment Program (UNEP, https://wesr.unep.org). The consumption of $CCl_4$ for dispersive uses in China was 0.20–0.26 Gg yr$^{-1}$ during 2010–2018 and 0.11–0.14 Gg yr$^{-1}$ during 2019–2021 (Fig. 2b). Similarly, the emission estimates of $CCl_4$ after 2010, as inferred from atmospheric observations (5.6–27 Gg yr$^{-1}$)[10–12,14], were higher than the reported consumption of $CCl_4$ (Fig. 2b). These results suggest (1) larger consumption of $CCl_4$ than that reported, (2) continuous emissions of $CCl_4$ from other sectors in addition to the reported consumption sectors, and (3) a combination of these two scenarios.

The northeastern region of Shandong, the eastern region of Jiangsu, and the southeastern region of Zhejiang were identified in this study as major source regions of $CCl_4$ in eastern China during 2021–2022 (Fig. 3; Supplementary Figs. 4 and 5). Additionally, air masses corresponding to high concentrations of $CCl_4$ typically pass through these regions (Supplementary Fig. 6, see Supplementary Discussion 2 for more details). The high emissions of $CCl_4$ from Shandong, Zhejiang, and Jiangsu were generally consistent with those reported by Park et al.[14] based on observation data from GSN, South Korea. However, our study also revealed emissions of $CCl_4$ from Jiangxi ($1.1 \pm 0.92$ Gg yr$^{-1}$) and Fujian ($0.99 \pm 0.64$ Gg yr$^{-1}$) provinces (Fig. 3 and Supplementary Table 1) that were not quantified by Park et al.[14]. This discrepancy is likely due to the GSN observations being less sensitive to emissions in these provinces (Fig. 1). Overall, a large amount of $CCl_4$ emissions were detected in eastern China, emphasizing the continued necessity to perform observations of $CCl_4$.

### Implications for ODS mitigations
The emissions of $CCl_4$ from emissive uses in China were estimated to be zero using the bottom-up inventory method by Fang et al.[7] and Wan et al.[21], as they set the consumption of $CCl_4$ for dispersive uses to zero. Although some bottom-up estimations of $CCl_4$ emissions included $CCl_4$ consumption for dispersive uses and other uses (e.g., feedstock), the emission estimates by Bie et al.[8] ($5.1 \pm 3.5$ Gg yr$^{-1}$ in 2014) and Sherry et al.[9] (7.3 Gg yr$^{-1}$ in 2014) were lower than the top-down estimates for 2014 ($26.5 \pm 8.1$ Gg yr$^{-1}$ by Park et al.[11] and $18.2 \pm 5.6$ Gg yr$^{-1}$ by Lunt et al.[12]). Of note, comparisons between bottom-up and top-down $CCl_4$ estimates after 2014 are not possible as post-2014 bottom-up emission estimates are lacking. Thus, further comprehensive bottom-up and top-down studies are strongly recommended to bridge the associated emission gaps for $CCl_4$.

The relative importance of $CCl_4$ emissions is increasing in China. For instance, the ozone depletion potential (ODP)-weighted emissions (equivalent to CFC-11 emissions) of $CCl_4$ in eastern China estimated in this study ($6.6 \pm 1.5$ CFC-11-eq Gg yr$^{-1}$) is comparable to the unexpected emissions of CFC-11 ($7.0 \pm 4.0$ Gg yr$^{-1}$) in eastern China during the 2014–2017 period relative to those during 2008–2012[14]. Since the CFC-

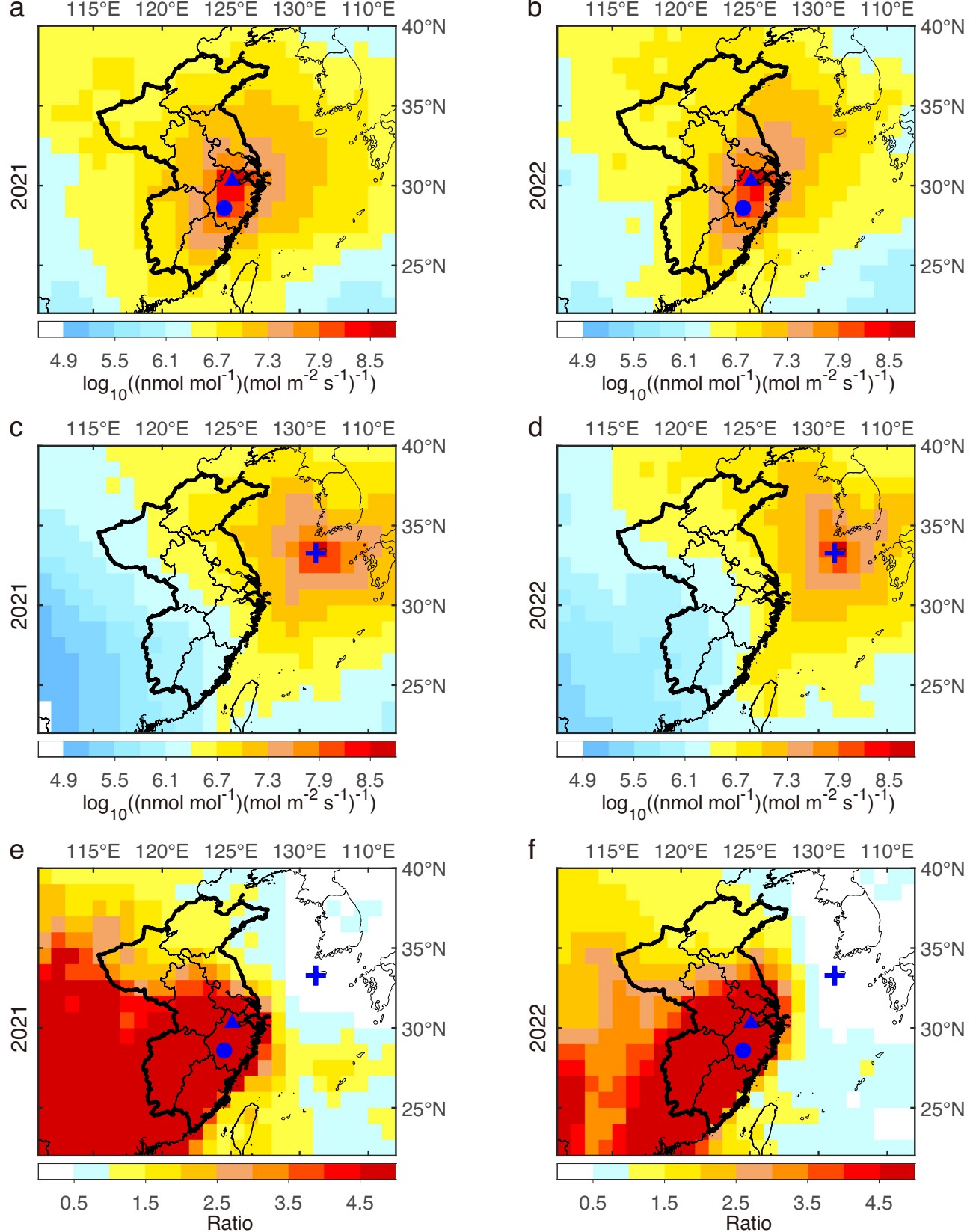

**Fig. 1 | Average emission sensitivity derived from FLEXPART simulations and ratios between different sites in 2021 and 2022. a, b** Average emission sensitivities for ZJU (blue triangle) and SHH (blue dot) sites. **c, d** for the GSN (blue cross) site. **e, f** Ratio of the average emission sensitivity of ZJU and SHH stations to that of the GSN site. The area framed by the bold black line is the target area (provincial names shown in Supplementary Fig. 1) for the inversion. ZJU, Zhejiang University; SHH, Shanghuang; GSN, Gosan. A larger domain is shown in Supplementary Fig. 2.

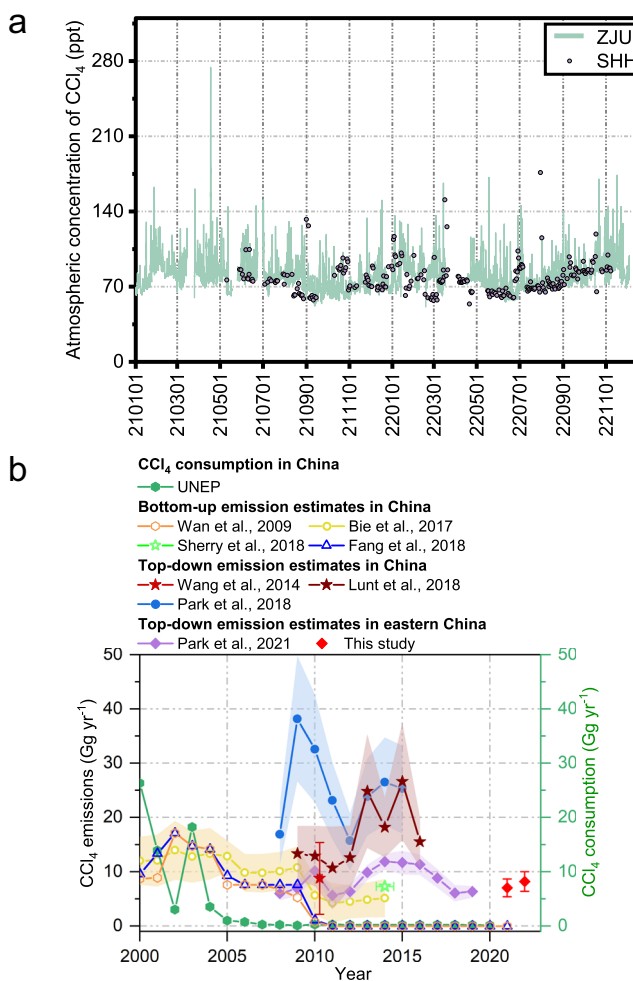

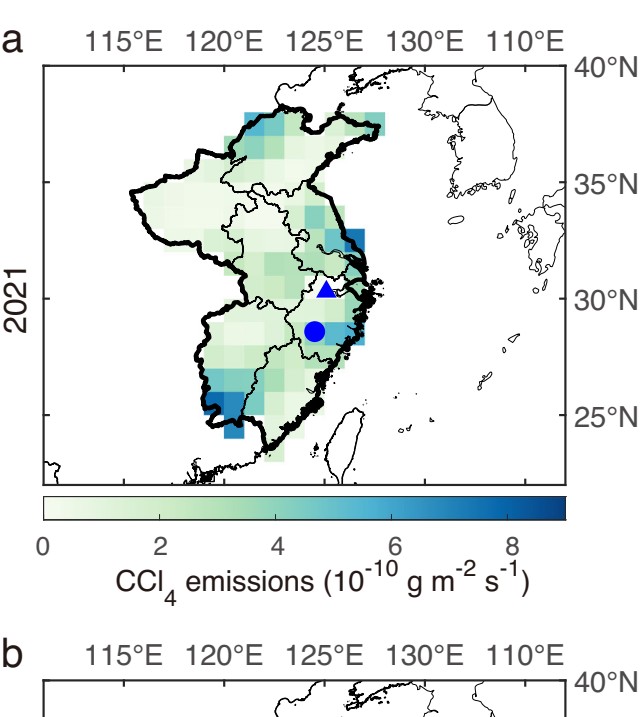

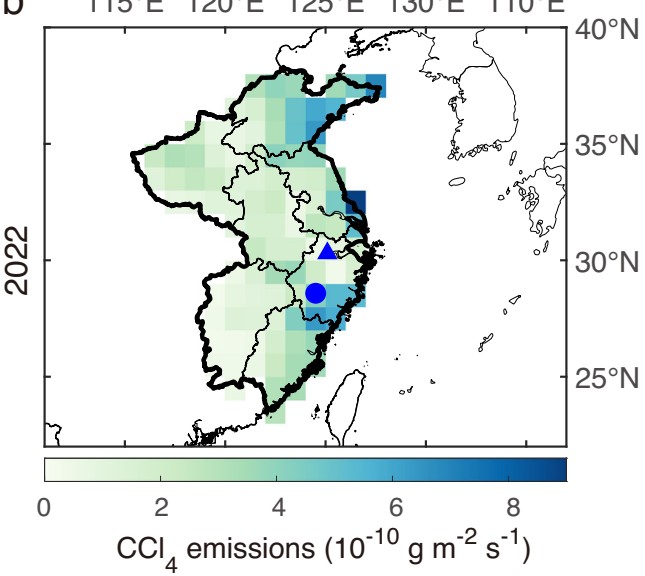

Fig. 2 | Concentration levels and emission estimates of CCl₄. a Time series of
CCl₄ concentrations at ZJU (cyan line) and SHH (lilac dot) sites during 2021–2022.
b Comparison of CCl₄ emission estimates in this study (red diamond with the error
bars representing ±1 standard deviation) with those from previous studies[7–12,21]. ZJU
Zhejiang University, SHH Shanghuang, UNEP United Nations Environment
Program.

Fig. 3 | Mean Spatial distribution of CCl₄ emissions in eastern China derived
from uniformly distributed prior emissions. a For 2021. b For 2022. Spatial dis-
tributions of CCl₄ emissions from each inversion using REBS or AGAGE baseline
filtering method are shown in Supplementary Fig. 4. Supplementary Fig. 5 is the
same as Supplementary Fig. 4 using population-proxy prior emissions. ZJU and SHH
sites are represented by the blue triangle and dot, respectively. REBS, robust
extraction of baseline signal method; AGAGE advanced global atmospheric gases
experiment, ZJU Zhejiang University, SHH Shanghuang.

11 emissions were reduced between 2017 and 2019, becoming lower
than the ODP-weighted CCl₄ emissions[14], emissions of CCl₄ have
become increasingly important in eastern China. Moreover, ODP-
weighted CCl₄ emissions in eastern China are several times higher than
the ODP-weighted emissions of other major ODSs, for example, it is
17.6 times higher than the 0.38 ± 0.75 CFC-11-eq Gg yr⁻¹ of dichlorodi-
fluoromethane (CFC-12) in 2019 in eastern China[14], 9.5 times the 0.69
(0.38–1.0) CFC-11-eq Gg yr⁻¹ of 1,1-dichloro-1-fluoroethane (HCFC-141b)
in 2020 in eastern China[22], and 1.4 times the 4.6 (3.9–5.3) CFC-11-eq Gg
yr⁻¹ of chlorodifluoromethane (HCFC-22; but for all of China) in 2019[23].
Therefore, this study concludes that further investigations on emis-
sions of CCl₄ and the continuation of atmospheric monitoring of CCl₄
in East Asia are necessary to preserve the ozone layer.

The emissions of CCl₄ in eastern China during 2021–2022 (7.6 ± 1.7
Gg yr⁻¹ in this study) were comparable to those before 2019 (6.1 ± 1.5 Gg
yr⁻¹ in 2018 and 6.3 ± 1.0 Gg yr⁻¹ in 2019)[16]. This comparison indicates
that the emissions of CCl₄ in eastern China did not fall over this period.
Based on the field measurement-based emission factors, Li et al.[24]
reported that CCl₄ emissions from chloromethane production plants
were only 2.2 ± 1.6 Gg yr⁻¹ in 2019 in China. The consumption of CCl₄
for dispersive uses as reported by China to the UNEP (https://wesr.
unep.org) was 0.12 Gg yr⁻¹ in 2021. Thus, the total emissions of CCl₄
from chloromethane production plants and dispersive uses in China

are smaller than our top-down emission estimates, indicating the
presence of other sources of CCl₄ emissions.

## Potential industrial sectors of CCl₄ emissions
A comprehensive measurement campaign comprising 456 exhaust
samples from 17 industry sectors was performed to determine
potential sources of CCl₄ (see Methods). This campaign shows that the
manufacturing of general-purpose machinery (MGPM, mainly manu-
facturing engines, excavators, and other heavy machinery; see Sup-
plementary Table 2) sector emits high concentrations of CCl₄. In the
MGPM sector, the highest concentration of CCl₄ was 103 ppb (parts
per billion, 1 ppb = 1000 ppt), and the average concentration was 5.4

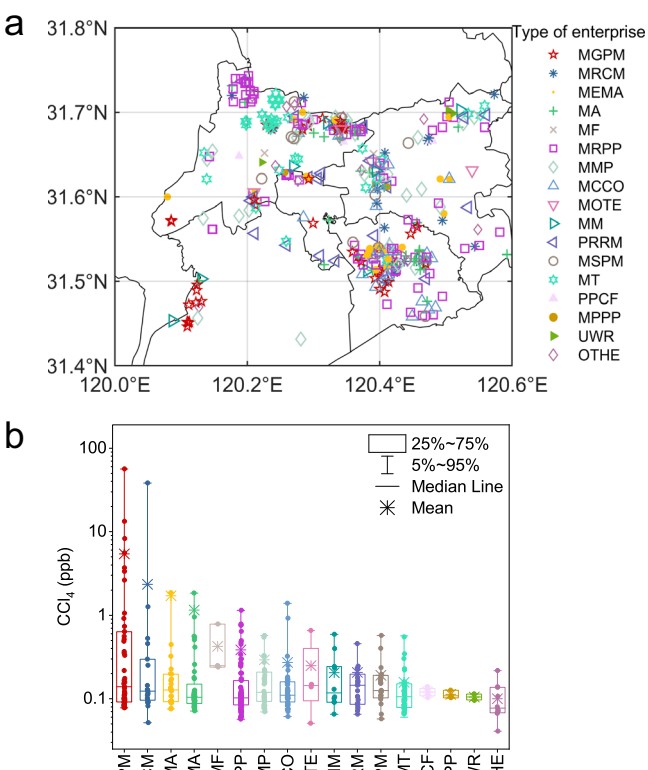

**Fig. 4 | Measured CCl₄ concentrations from each industry sector. a** Spatial distribution of sampling industry sectors with different symbols and colors. **b** Measured CCl₄ concentrations from each industry sector. Box plots indicate the median (middle line), 25th, 75th percentile (box), and 5th and 95th percentile (whiskers) as well as the mean level (asterisks). Solid dots represent the concentration of each sample. MGPM manufacture of general-purpose machinery, MRCM manufacture of raw chemical materials and chemical products, MEMA manufacture of electrical machinery and apparatus, MA manufacture of automobiles, MF manufacture of paper and paper products, MRPP manufacture of rubber and plastics products, MSPM manufacture of special purpose machinery, MT manufacture of textile, PPCF processing of petroleum, coal, and other fuels, MPPP manufacture of paper and paper products, UWR utilization of waste resources, OTHE others.

ppb (Fig. 4). For the manufacture of raw chemical materials and chemical products (MRCM, mainly manufacturing coatings and resins) sector, the concentration of CCl₄ reached 38 ppb, with an average concentration of 2.3 ppb. Moreover, the manufacture of electrical machinery and apparatus (MEMA) and manufacture of automobiles (MA) sectors had average CCl₄ concentrations of 1.8 ppb and 1.2 ppb, respectively. The average concentrations of CCl₄ in MGPM, MRCM, MEMA, and MA sectors were approximately 71, 30, 24, and 16 times the northern hemispheric atmospheric background concentration (~ 0.076 ppb, http://agage.mit.edu/), respectively, and 67, 29, 23, and 15 times the average concentration of CCl₄ at ZJU and SHH (~ 0.080 ppb), respectively.

The high concentrations of CCl₄ in these sectors may arise from (1) byproducts of CCl₄ during the use or production of other chemicals/products, (2) using CCl₄ as feedstock to produce other chemicals/products, (3) uses of industrial products with impurity of CCl₄ (e.g., CCl₄ in chloroform[17]), or (4) other unidentified uses of CCl₄. Of note, the first three processes mentioned above are not controlled by the Montreal Protocol, and countries with these processes if any are not against the Montreal Protocol. Nevertheless, our observations reveal that the MGPM, MRCM, MEMA, and MA sectors are potential sources of CCl₄ emissions and have not been identified as such in previous

studies nor under the on-site enterprise supervision of CCl₄. The average concentrations of CCl₄ from other sectors (see Supplementary Table 2 for detailed information) sampled in this study were between 0.09 and 0.42 ppb (Fig. 4), only slightly higher than the atmospheric background level (~ 0.076 ppb) and were, therefore, not deemed to be as important as the MGPM, MRCM, MEMA, and MA sectors.

The number of enterprises has exhibited an obvious upward trend for the MGPM and MEMA industries since 2018 (growth rate: 10–16% yr⁻¹ during 2018–2021)[25] (Supplementary Fig. 7). Although the number of enterprises in most other industries was relatively stable during 2011–2021[26] (Supplementary Fig. 7), the overall trend has been on the rise (Supplementary Fig. 7). Considering that they may represent potential sources of CCl₄ emissions, attention should be paid to these industries (particularly MGPM) to evaluate their impact on the ozone layer.

This study quantified substantial emissions of CCl₄ in eastern China (7.6 ± 1.7 Gg yr⁻¹) during 2021–2022 based on observations at two sites located in Zhejiang Province, China. By conducting extensive sampling, we detected high concentrations of CCl₄ in the exhaust gases from the manufacture of general-purpose machinery and the manufacture of raw chemical materials, and chemical products, implying that these industrial sectors may be potential sources of CCl₄. These findings strongly endorse continued monitoring of atmospheric CCl₄ to protect the ozone layer.

## Methods

### Observations of atmospheric CCl₄ in eastern China

At the ZJU site, in-situ hourly samples were collected for 30 min approximately 20 m above the ground between January 2021 and December 2022. In total, 11,174 samples were collected at the ZJU site over two years. A total of 294 samples were collected at the SHH site. Samples at SHH were collected in 3.2-L stainless steel canisters (Entech Instrument, Inc., Simi Valley, CA, USA) at approximately 1100 masl for 1 min at 14:00 LT (local time) between May 2021 and November 2022 (no sampling in case of rainy or snowy days) and transported to the Zhejiang University laboratory for analysis. There were no apparent CCl₄ emission sources in the immediate regions near the ZJU or SHH sites.

Samples from both sites were quantitatively analyzed by gas chromatography coupled with a mass spectrometry detector (GC-MSD; ZF-PKU-VOC1007, Beijing, China). Briefly, after removing H₂O and carbon dioxide (CO₂), the sample was enriched in a cold trap at −160 °C, then desorbed at 120 °C for 3 min. The desorbed sample was separated in the GC system equipped with a DB-624 column (60 m length × 0.25 mm i.d. × 1.4 μm film thickness; Agilent Technology, Santa Clara, CA, USA) with helium as the carrier gas. The standard gas used for determining CCl₄ was TO-15 (Linde Gas North America LLC, Medford, OR, USA). The linear coefficient (expressed as $R^2$) of the five-point calibration curve for CCl₄ was 0.998, and the detection limit was 14 ppt (parts per trillion), ~5.5 times lower than the average measured concentrations of CCl₄ (80.0 ± 13.6 ppt at ZJU and 77.5 ± 15.3 ppt at SHH).

### Inverse modeling of CCl₄ emissions in eastern China

This study used an inverse modeling technique based on a FLEXPART atmospheric transport simulation model and the Bayesian inversion algorithm to quantify CCl₄ emissions. The FLEXPART-based inversion method has been applied in many previous studies[27–29]. In brief, driven by the meteorological data (European Center for Medium-Range Weather Forecasts) with a spatial resolution of 1° × 1° and a temporal resolution of 3 h, the FLEXPART model was run in backward mode for 20 days. The source–receptor relationship (termed as "emission sensitivity") matrix was established based on the backward simulation. Combining the derived emission sensitivity matrix, Bayesian inversion technique, and the CCl₄ data from ZJU and SHH, yielded the CCl₄

emission strength (1° × 1°) in grid cells over eastern China. The associated equations are as follows:

$$J(x) = \frac{1}{2}(\mathbf{x} - \mathbf{x_a})^T \mathbf{S_a}^{-1}(\mathbf{x} - \mathbf{x_a}) + \frac{1}{2}(\mathbf{y}^{obs} - \mathbf{Hx})^T \mathbf{S_o}^{-1}(\mathbf{y}^{obs} - \mathbf{Hx}) \quad (1)$$

By solving $\nabla_x J(x) = 0$ yields:

$$\mathbf{x} = \mathbf{x_a} + \mathbf{S_a}\mathbf{H}^T(\mathbf{HS_aH}^T + \mathbf{S_o})^{-1}(\mathbf{y}^{obs} - \mathbf{Hx_a}) \quad (2)$$

$$\mathbf{S_b} = (\mathbf{H}^T\mathbf{S_o}^{-1}\mathbf{H} + \mathbf{S_a}^{-1})^{-1} \quad (3)$$

where $\mathbf{x}$ represents the state vector of the emission strength, $\mathbf{y}^{obs}$ represents the $CCl_4$ measurement vector, $\mathbf{x_a}$ represents the prior emission vector, $\mathbf{H}$ is the emission sensitivity, $\mathbf{S_a}$ and $\mathbf{S_b}$ are the error covariance matrix of prior and posterior emissions, respectively, and $\mathbf{S_o}$ represents the error covariance matrix of measurement data. To obtain the prior emission vector ($\mathbf{x_a}$), the national total $CCl_4$ emissions in China during 2011–2015 ($23.6 \pm 7.1$ Gg yr$^{-1}$)[11] and emissions in other countries (derived from global emissions [$44 \pm 15$ Gg yr$^{-1}$ in 2020][1] and emission estimates of the United States [4.0 (2.0–6.5) Gg yr$^{-1}$ during 2008–2012][30], Japan [0.6 Gg yr$^{-1}$ in 2014][9], South Korea [0.2 Gg yr$^{-1}$ in 2014][9], and India [2.8 Gg yr$^{-1}$ in 2014][9]) were assigned to grid cells with a uniform spatial distribution in eastern China, and a population-proxy distribution in other regions (Supplementary Fig. 8a). The inversion results were also evaluated using prior emissions following population distribution in 2020[31] (Supplementary Fig. 8b).

The robust extraction of baseline signal (REBS) method[32] was applied to distinguish background and non-background concentrations of the in-situ $CCl_4$ concentrations at ZJU. The REBS is a statistical method developed by Ruckstuhl et al.[32] to extract background signals using a robust local regression model and has been widely applied to determine baselines of trace gases in inversion studies[22,33]. The observed concentration at a certain time ($y(t_i)$) was divided into three parts as depicted in the following equation:

$$y(t_i) = g(t_i) + m(t_i) + e_i \quad (4)$$

where $g(t_i)$ represents the background concentration at time $t_i$, $m(t_i)$ is the enhanced concentration caused by polluted plum during $t_i$, and $e_i$ represents the observational error.

The baseline curve g was obtained using the REBS technique over a sufficiently long bandwidth (90 days) by assuming that most observations are at background levels and that the baseline signal changes slowly relative to the regional signal. In this method, data points closer to the time of consideration were given more weight, and data points outside a specific range (1.5σ in this study) were iteratively excluded.

For flask samples at SHH, the background concentration of $CCl_4$ was determined as the lowest concentration measured in a two-month moving window. The observational error ($\sigma_{obs}$) was calculated as follows:

$$\sigma_{obs} = \sqrt{\sigma_{obs\_precision}^2 + \sigma_{obs\_representation}^2 + \sigma_{background}^2} \quad (5)$$

where $\sigma_{obs\_precision}$ is the measurement precision of $CCl_4$, $\sigma_{obs\_representation}$ stands for representation of the observation, $\sigma_{background}$ represents the background uncertainty. In this study, $\sigma_{obs\_representation}$ was calculated as 1-Sigma standard deviation of the measurements each day for online samples and as 1-Sigma standard deviation of the measurements in a two-month moving window for flask samples. The $\sigma_{background}$ was estimated as 1-Sigma standard deviation of the fitted background concentrations during the sampling period. The diagonal elements of $\mathbf{S_o}$ were set to squared $\sigma_{obs}$.

The diagonal elements of $\mathbf{S_a}$ were calculated as squared uncertainty of the prior emission field. The off-diagonal elements of $\mathbf{S_a}$ were calculated based on previous studies[29,34], with the spatial decorrelation length scale set to 400 km. In this study, three sets of prior emissions were established (i.e., 150%, 100%, and 50% of the reference prior emissions), and three uncertainties (600%, 450%, and 300% for uniformly distributed prior emissions) were set under each set of prior emissions. A total of nine inversions were carried out for each year. The final posterior emissions were the average of the nine inversions (Supplementary Fig. 9). The posterior $CCl_4$ emissions in eastern China under the nine inversions in 2021 and 2022 were loaded in the range of the three sets of prior emissions (Supplementary Fig. 9). The posterior emissions were insensitive (varied <9%) to variations in emission uncertainty, suggesting that the prior emissions used in this study were not systematically high or low, and that the observations constrain posterior emissions well. We also tested baselines from the Advanced Global Atmospheric Gases Experiment (AGAGE) baseline filtering method (baseline obtained by fitting the daily minimum with a second-order polynomial)[35] to examine the impact of different baselines on posterior emissions. The inversion ensembles (Supplementary Table 3) using two baselines (AGAGE and REBS methods) and two prior emission fields (uniform and population-proxy distributions) show that the posterior emissions varied by <6% (standard deviation divided by mean) during 2021–2022 (Supplementary Fig. 10).

### Field sampling and measurements of $CCl_4$ among industrial sectors

A comprehensive sampling campaign of 379 industrial enterprises was carried out from September 2021 to January 2022 in a city in eastern China (Fig. 4a; the number of samples in each industry sector is shown in Supplementary Table 2). Finally, 459 valid samples were obtained from the chimney vents of the target enterprises (see sampling method details in Supplementary Information). The industrial samples collected in this study were analyzed using the same analytical system as the ZJU and SHH samples.

## Data availability

The emission sensitivity data sets generated in this study have been deposited in the Figshare (https://doi.org/10.6084/m9.figshare.24499582). Other data supporting the findings in this work are available within the manuscript and Supplementary Information file and available from the corresponding authors upon request.

## Code availability

The code of the dispersion model FLEXPART is available from https://www.flexpart.eu. The code for the FLEXPART-based Bayesian inversion is available on request from X.F.

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

## Acknowledgements

The authors are grateful for the support provided by the Key R&D Program of Zhejiang Province (No. 2022C03154 to X.F.), the National Key Research and Development Program of China (No. 2019YFC0214500 to X.F.), and the National Natural Science Foundation of China (No. 22106134 to X.F.).

## Author contributions

X.F. designed the research. B.L., J.H., L.Z., and X.L. conducted the sampling campaign. B.L. performed the inverse modeling, drew the graphs, and organized the data. J.H., L.Z., X.L., X.H., M.M., L.H., D.C., Q.D., Y.S., Z.C., A. C., R.F., and R.P. revised the manuscript.

## Competing interests

The authors declare no competing interests.
