## [Peer Review File NEW · Nature Communications]

Substantial CCl₄ emissions in eastern China during 2021–2022 and exploration of potential new sourcesEditorial Note: Parts of this Peer Review File have been redacted as indicated to remove third-party material where no permission to publish could be obtained.

Reviewer #1 (Remarks to the Author):

Please see attached PDF file.

Reviewer #1 Attachment on the following page

Review of NCOMMS-23-29246, Substantial carbon tetrachloride emissions in eastern China, after 2010 dispersive-use phase-out and 2019–2022 strict government supervision: A concern for the ozone layer, by Li et al.

I shall first briefly respond first to the questions posed on Authors instructions, then expand.

What are the noteworthy results?

By far, the most noteworthy results are the measurements of the content of CCl₄ in the effluent of 17 types of industries, as detailed mainly in the Supplement and described at the end of the paper.

The authors have mainly highlighted their estimate of the emission of CCl₄ for Eastern China as a noteworthy result; in my opinion, several aspects of the calculation of the emission of CCl₄ for Eastern China are deeply flawed and the uncertainty value for this emission is unjustifiably low.

Will the work be of significance to the field and related fields? How does it compare to the established literature? If the work is not original, please provide relevant references.

The aspect of this work I think is most noteworthy, described just above, is very important, new and unique, and would have very strong impact. The aspect of the work the authors have chosen to emphasize is not, in my opinion, either new enough or strongly enough quantified to warrant publication in *Nature Communications*.

Does the work support the conclusions and claims, or is additional evidence needed?

No for the aspect of the work the authors have emphasized (emission of CCl₄ for Eastern China); yes for the aspect I feel is most important (emission of CCl₄ from various industries).

Are there any flaws in the data analysis, interpretation and conclusions? Do these prohibit publication or require revision?

Yes, as I will elaborate below. Simply, using the nation-wide estimate of CHCl₃ for two years, 2013 and 2015 (Supplemental Table 1) is flawed because the target gas, CCl₄, does not exhibit a strong enough correlation with CHCl₃ to justify the use of CHCl₃ as a tracer for the emission of CCl₄. Also, a lot has happened in China since 2015, following the 2018 publication of the Montzka et al. paper. Applying data from 2013 and 2015 to infer emissions in 2020 and 2021 is not sound. Finally and very importantly, the data could be used in an inverse model framework to compute emissions of CCl₄ from Eastern China. Yet this approach is not used.

Is the methodology sound? Does the work meet the expected standards in your field?

Methods not sound, as noted above.

In terms of expected standards in the field, there are a large number of problems including the reader being asked to believe there is no substantial bias between the standard used for Trinidad Head Station (the Scripps Institution of Oceanography USA25 standard) and the standard used at their station (the Linde Gas standard), with reference only in the literature and no support for supplement. In papers on this topic where mixed standards are used, it is

common to report side-by-side measurements of a sample from a **current** lab capable of running both standards, to quantify the actual bias).

The quantification of the emission of CCl₄ from Eastern China is well below the expected standard, which is to make use of an inverse (or perhaps data assimilation) model.

Is there enough detail provided in the methods for the work to be reproduced?

No. There is minimal discussion of either the Robust Extraction of Baseline Signals (REBS) or the trajectory clustering method in either Main, Methods, or Supplement.

Now I will summarize the paper and highlight my “Major” and “Minor” points.

Overview:

First off, the paper is **exceptionally well-written**. Other than the occasional use of “This” or “These” as a noun, there is little I can offer to improve the communication of the message. I sincerely appreciate being sent a paper to review that is so well-written.

The paper reports observations of CCl₄ and related species from two sites in Eastern China. The observations are stated to be hourly; 11,174 sample were acquired at one site between Jan 2021 and Dec 2022 whereas 294 samples were acquired at the other site between May 2021 and Nov 2022. For the site with the larger number of samples, that were collected for about 65% of the sampling opportunities (that is, 11,1174 / 2x8760). For the other site, data were collected for only about 2.5% of the sampling opportunities (this is, 294 / 16/12 x 8760)

The data are used to compared mean and various percentiles of CCl₄ to a time series from the Trinidad Head station, which reveals frequent enhancements of CCl₄ in Eastern China and to compute the emission of CCl₄ in Eastern China in 2020 and 2021. Finally, observations of CCl₄ are reported from the exhaust of 17 types of enterprises in Wuxi, which is in the Jiangsu province of Eastern China. These observations show highly elevated levels of CCl₄ in the following sections: Manufacture of General Purpose Machinery mainly manufacturing of engines, excavators, and other heavy machinery); Manufacture of Raw Chemical Materials and Chemical Products industry (mainly manufacturing coatings and resins), Manufacture of Electrical Machinery and Apparatus, and the Manufacture of Automobiles.

In my opinion, the comparison of time series for CCl₄ to data from Trinidad Head (that is, Figure 1b) is likely not enough to warrant publication in *Nature Communications*, because the contemporary emission of CCl₄ from Eastern China is simply well known. The only possible surprise here would have been the disappearance of elevated CCl₄ from Eastern China. I state the elevation of CCl₄ in Eastern China is well known based on the content of numerous other studies cited in the paper and that have data which are used in Figure 1b, as well as a very important additional paper (Benish, S. E. et al., Airborne observations of CFCs over Hebei Province, China in Spring, JGR, 126, e2021JD035152, 2021) that is not cited (lots more on the importance of this paper below.

In my opinion, the computation of the emission of CCl₄ for Eastern China in *Nature Communications* is also not warranted, because CCl₄ exhibits such a poor correlation with CHCl₃, the gas selected as the “Tracer” for Equation 4. Figure 2 of the Supplement shows

almost no correlation between these two species; the data look like a traditional “shotgun” blast, with the slope clearly influenced by a single outlier. Um, under what conditions was this outlier obtained? And, how does the slope change with the neglect of this single outlier? Any result that depends on a single outlier is highly suspicious. Finally and most importantly, how can the slope of the data (“0.087” with an uncertainty of *only* “0.001”) be known so well, when the data (outside this one outlier) exhibit almost no correlation? Please note Benish et al. (their Figure 4) report a value of r^2 between CCl₄ and CHCl₃ of only 0.03 based on airborne sampling over the Hebei Province in spring of 2016. A much higher value of r^2 of 0.78 is reported between CCl₄ and CFC-11.

Finally, in my opinion, the most important part of this paper is not given due justice. I have extracted from the **Potential industrial sectors of CCl₄ emissions** section of the paper the four dominant industrial sectors. None of these sectors are named in the abstract, which only states:

Subsequently, our study identified potential industrial sources of CCl₄ emissions that were not revealed in previous studies and not controlled by the governmental supervision of CCl₄. Therefore, our study can guide mitigations of CCl₄ emission and endorse continued monitoring of atmospheric CCl₄ to protect the ozone layer.

I shall provide further detail on a few Major and Minor points below.

Major Points

1) The numerical values of the emission of CCl₄ for Eastern China in 2021 and 2022 are highly suspicious, due to the use of CHCl₃ as “the tracer”, the use of emissions for CHCl₃ from much earlier years (2013 & 2015), and the report by others (that is, Benish et al. 2021) of much higher correlations between CCl₄ and other species, such as CFC-11. If the authors see a high correlation of CCl₄ with CFC-11 that is not dependent on a single outlier, then perhaps CFC-11 can be used as “the tracer”. Regardless, whatever tracer, if the authors insist on following this approach, upon revision I suggest they:

- a) describe the correlation of the tracer with CCl₄
- b) state how the uncertainty in the slope is found
- c) justify the use of much older emission estimates (for CHCl₃, for 2013 and 2015) to a more recent time period, given so much attention to the emission of ODSs in China following the publication in 2018 of the Montzka et al. paper.
- d) Most importantly show that the selected tracer exhibits a meaningful correlation with CCl₄, not only for the ~2 year station data, but also for the samples obtained in the exhaust of 17 specific industries. If CHCl₃ show a tight correlation for data from Manufacture of General Purpose Machinery, Raw Chemical Materials and Chemical Products, Electrical Machinery and Apparatus, and Automobiles, then perhaps this gas can be used as a suitable tracer. The submitted paper does not describe any other data for the 17 specific industries, other than CCl₄. If there is little to no correlation between CHCl₃ and CCl₄, then another tracer is needed. Regardless, the selected tracer **MUST** show a significant correlation with CCl₄ for both the station data and the exhaust data, for the suspicion noted above to be assuaged.

Another option I encourage the authors to consider, should they insist on attempting to quantify the emission of CCl₄ for Eastern China in 2021 and 2022, is to use either the inverse model approach or perhaps data assimilation.

One more option, one I prefer, is that if this paper proceeds in *Nature Communications*, the finding of significant emissions of CCl₄ from certain industrial sectors be the primary point of emphasis. I expand further on this suggestion in Major Point #4, below.

2) I suggest the authors provide better justification of the combination of data from Trinidad Head Station obtained using the Scripps Institution of Oceanography USA25 standard with data from their stations, that use the Linde Gas standard. Seeing a comparison of samples run through a lab with one standard, against data measured using the other standard, is common (for example, Tables S2, S3, and S4 of the previously mentioned Benish et al. paper). Also, upon revision, there are many places where the results of Benish et al. 2021 should be cited, and *data from this paper should be added to Figure 2b*.

3) As noted above, there is minimal description of either the Robust Extraction of Baseline Signals (REBS) or the trajectory clustering method in either Main, Methods, or Supplement. Upon revision, more details are needed. For the trajectory clustering method, the HYSPLIT model is used, yet there are no references for HYSPLIT apparent in the submitted manuscript. I suggest citing this paper by Stein et al. <https://doi.org/10.1175/BAMS-D-14-00110.1>. Also, Figure 1 of Supplement is hard to understand. It is not clear which colors correspond to which numbered trajectory, and the use of yellow (for SHH) is ill-advised, as this line is very hard to see. I also suggest writing “ZJU” onto both panels of Fig 1a and “SHH” on both panels of Fig 1b.

4) As noted above, I feel the most important result of this paper is not highlighted adequately. If this paper proceeds in Nature Communication, I suggest dispensing with the comparison of the new measurements of CCl₄ in 2020 and 2021 with data from Trinidad Head quickly, either improving or perhaps striking the attempt to compute emissions of CCl₄ for Eastern China over 2020 and 2021, then moving a lot of the material in Supplement regarding what I believe is the **key new message** of this paper into Main (or Methods).

Minor points

a) The ODP and GWP for CCl₄ given in the most recent Scientific Assessment Ozone Depletion report are 0.87 and 2150, respectively. Upon revision, please adopt these values (since they are current and have been assessed by the relevant experts) on lines 38 and 39, and in the rest of the paper (that is, CFC-11 equivalent emissions) that rely on the use of these numbers.

b) Line 50. Again, as noted above, the paper is extremely well written! Minor point, but rather than “to restrict CCl₄ emissions”, I suggest “to understand the reason for the gap in bottom-up and top-down emissions”. Regardless, if the paper is revised as suggested above, there will certainly be much new text regarding how the observations will be used in the future “to restrict emissions”.

c) Benish et al., multiple lines: Benish et al. could also be cited on line 68 as well as lines 76 to 79. : Benish et al. also showed, based on their strong correlation of CFC-11 with CCl₄, the likely co-production of these two species (line 82).

- d) Line 81. “observed” should be “reported”, since the paper was published in 1981.
- e) Lines 84, 86, and 88. The use of “16” on lines 84 & 86 and “19” on line 88 is confusing. If these numbers are truly different, a bit more explanation would be helpful.
- f) Line 95. Suggest “this study describes observations of ...”
- g) Line 99. Here, I prefer “substantial emissions of CCl₄” since substantial modifies emissions
- h) Line 121. Not sure “41N” is really “close” to 30.306 and 28.583N; perhaps a different word can be used
- i) Line 125. Suggest “These observations indicate ...”
- j) Line 131. In addition to the request for more information about REBS, the inclusion of a reference here would be helpful
- k) Line 135. Suggest “respectively” rather than “in respect”
- l) Lines 146 to 149. Not sure all of this detail is needed, especially if the paper’s focus is altered, but if this text survives, should also cite numerical values from the Benish et al. study
- m) Lines 153 to 156. The use of HYSPLIT for the trajectory clustering should be stated, with a reference to a HYSPLIT paper, perhaps the above mentioned Stein et al. study
- n) Lines 164 to 165. A citation for the CWT method should be given
- o) Line 235. Suggest “This comparison indicates ... did not fall over this period of time” although, given the inherent problems I have identified with the estimate for the computation of the emission of CCl₄ in Eastern China, I have to also state this sentence may need to be altered in a more significant manner, or perhaps removed.
- p) Line 268. Suggest “Our observations reveal ...”
- q) Line 307. In addition to the website, should also provide a reference here, for HYSPLIT.
- r) Line 506. Figure 1 is very important: this figure is the first graphical component of the paper. The figure should be improved, either by zooming in to make the geographic region of China stand out more clearly, or perhaps some other manner. The three words on panel c are hard to read. Finally, I suggest also pointing out the location of Beijing, so that readers can better orient the locations of ZJU, SHH, and Gossan.
- s) Line 518. I have stared at this figure, and can not find either the black or pink dots. If these dots are truly present, they should be made more visible.
- t) Line 527. Figure 5c is very very important, in my opinion. Figure 5b is easy to “understand”, but hard to decipher, as many of the colors are similar. Perhaps various symbols with a smaller number of colors should be used. Finally, Figure 5a is hard to understand: what is the inset?

END OF REVIEW

Reviewer #2 (Remarks to the Author):

This manuscript reports on unique measurements of the ozone-depleting CCl₄ from two new sites in China during 2021 and 2022. Quantifying CCl₄ emissions from atmospheric measurements is an important issue worth of consideration by this journal because substantial CCl₄ emissions persist globally and are delaying recovery of the ozone layer, but the sources responsible for these emissions have not been adequately identified and characterized. Ongoing measurements from these new sites have the potential to inform us more accurately about emissions from unique regions of China than has been possible previously. The authors find that emissions from eastern China are comparable in magnitude to those published elsewhere for this same region through 2019 using measurement data from a different site (Gosan, S. Korea). The authors argue that the emissions they have derived for CCl₄ during 2021-2022 are unusually high owing to the "implementation of strict government supervision for CCl₄" after 2018. In an intriguing second part of the paper, the authors report on measurements of CCl₄ from air collected from the chimney vents of 379 businesses that represented 17 different industry sectors. Unusually high CCl₄ concentrations were observed from some sectors that haven't been associated with CCl₄ emissions in the past. While the data presented are unique and relevant to the issue of ozone depletion and ozone layer recovery, I cannot recommend publication at this time for a number of reasons that I delineate below.

First, the importance and uniqueness of this manuscript primarily relates to the availability of new measurement data from new sites AND the assertion that the derived emissions from eastern China are higher than they should be owing to the "implementation of strict government supervision for CCl₄" during this period. However, there is no indication that the purpose of the "supervision program" was to limit CCl₄ emissions. Certainly, it was instituted to improve tracking of produced CCl₄ so that large amounts couldn't be diverted to produce CFCs, but no indication is provided by the authors to suggest that this program might have had as its goal substantial reductions in CCl₄ emission (and I don't know this independently).

Second, the emission derivation is performed here in a manner that substantially underestimates the true uncertainties, making me question the reliability of the CCl₄ emission estimates provided. Emissions of CCl₄ in this work are derived by scaling the concentration enhancements for CCl₄ to those measured concurrently for CHCl₃ and multiplying that ratio by the "known" CHCl₃ emission. This method works best if the enhancements for the two gases are highly correlated. Figure 2 of the Supplementary material shows the correlation, but its significance isn't quantified or mentioned (and visually does not look very high). Most worrisome, however, is that emissions of CHCl₃ from eastern China from 2013 (48.3 Gg) and 2015 (80-95 Gg) (not 2021 or 2022) are used to derive emissions for CCl₄ in 2021 and 2022 based on measurements in this latter years. No consideration of what CHCl₃ emissions might have been in 2021 or 2022 is mentioned, making the estimate of CCl₄ emissions in the key years highly suspect, particularly since the large difference (change?) in CHCl₃ emissions during 2013 and 2015 suggests that CHCl₃ emissions from eastern China can change substantially from year-to-year.

Third, the authors also assert that they have found CCl₄ emissions in new regions of China (Jiangxi and Fujian provinces). They also acknowledge that the trajectory method can often erroneously place emissions upwind of the actual source region (my interpretation of the "ghost" sources in the ocean discussed in Figure 3). This seems also a possibility for the proposed CCl₄ emissions these two regions and would need further investigation before one could reliably assert that there are significant emissions coming from these provinces.

Fourth, in a highly unique section of the paper, concentration measurements were made from the stacks of many industries and potentially suggest the discovery of previously unrecognized CCl₄ sources. However, this result needs some further development before such assertions can be made with any reliability. Questions needing to be explored so that the implications of the new results can be better understood include: how extensive is this industry and how many businesses related to this industry exist in China? What are the potential fluxes of CCl₄ from these stacks? Can you put those two numbers together to estimate potential emissions from this sector? What are the potential underlying variables affecting the range of CCl₄ concentration in stacks associated with the same industry? Particularly MGPM? For this portion of the paper to be useful to a wider

audience, it will require some further estimation of these quantities, or reasonable limits applied to them, so that the potential importance of these other sources can be roughly assessed.

Reviewer #3 (Remarks to the Author):

Review of the manuscript: Substantial carbon tetrachloride emissions in eastern China, after 2010 dispersive-use phase out and 2019–2022 strict government supervision: A concern for the ozone layer by Li et al., submitted to Nature Comm.

The manuscript is related to on-going emissions of CCl₄, for which a consistent gap between bottom-up and top-down emissions exists globally, with higher emissions estimated by measurement-based top-down methods. The manuscript discusses enhanced CCl₄ concentrations in eastern China and estimates emissions, which are higher than expected. Authors also make use of industrial samples, which show massively high concentrations of CCl₄ and related them to certain types of industries.

I am in favour of publishing this manuscript in Nature Comm. because of its global implications. However, some of the comments should be covered carefully.

General comments

1. The most important finding is the massive concentrations close to industrial sites. Here an explanation with a high grade of probability is forbidden emissive uses of CCl₄. Authors should clearly state this in the manuscript. This would be a very good point to make and this would also close the gap between the bottom-up and the top-down estimate, as in the bottom-up numbers such illegal practices are simply not taken into account as it was deemed highly unlikely, but obviously not.
2. Accordingly, point number 1 should also be mentioned in the abstract. At the moment the paper stays very vague on this. The readers want to know the reason for these emissions and it does not make it better to hide it away.
3. Several citations are at the edge of misquotation. These will be detailed in the specific comments below.
4. Data must be openly accessible.

Specific comments:

Line 21

...was globally phased out by 2010, including China.

Line 23

...were disclosed, and the latest...

Line 27

...respectively, well above the global background.

Line 29

In 2010 and after the implementation...

Line 42

Feedstock instead of "raw material"

Line 42

Ref 6 is not needed for this argument and does not really cover this

Line 45

[8] is wrongly cited. In the report they say:

Global CCl₄ emissions did not significantly decline during the 2010–2019 period

(0.1 ± 0.2 Gg yr⁻¹),

41 ± 3 and 38 ± 3 are not mentioned in the report (or at least this reviewer has not seen it)

Line 47

Take the newest numbers from the most recent ozone assessment (2022)

Line 54

...CCl₄ emissions from dispersive uses were reduced...

Line 55

A bottom-up inventory for non-emissive uses of CCl₄, including...

Line 59

Moreover, another independent study showed bottom-up emission estimates...

Line 81

Rigby et al say at least 40-60%

Line 178

Equation 4 (see methods)

Line 187

Reference missing for REBS

Line 188

...for total China reported...

Line 203

...from emissive uses...

Line 214

The relative importance...

Line 375

Data have to be open access

Responses to Reviewer #1:

Review of NCOMMS-23-29246, Substantial carbon tetrachloride emissions in eastern China, after 2010 dispersive-use phase-out and 2019–2022 strict government supervision: A concern for the ozone layer, by Li et al.

I shall first briefly respond first to the questions posed on Authors instructions, then expand.

What are the noteworthy results?

By far, the most noteworthy results are the measurements of the content of CCl_4 in the effluent of 17 types of industries, as detailed mainly in the Supplement and described at the end of the paper.

The authors have mainly highlighted their estimate of the emission of CCl_4 for Eastern China as a noteworthy result; in my opinion, several aspects of the calculation of the emission of CCl_4 for Eastern China are deeply flawed and the uncertainty value for this emission is unjustifiably low.

Response:

Following the suggestions by reviewers, we have given up the interspecies correlation emission estimates and used emission inverse modeling (review comments suggest the method) in our revised manuscript. Thus, the flaws have been avoided, and the uncertainty value for this emission is justified (please see the detailed responses below).

Will the work be of significance to the field and related fields? How does it compare to the established literature? If the work is not original, please provide relevant references.

The aspect of this work I think is most noteworthy, described just above, is very important, new and unique, and would have very strong impact. The aspect of the work the authors have chosen to emphasize is not, in my opinion, either new enough or strongly enough quantified to warrant publication in Nature Communications.

Response:

We have improved our work and tried to emphasize the new and important aspects (please see the detailed responses below).

Does the work support the conclusions and claims, or is additional evidence needed?

No for the aspect of the work the authors have emphasized (emission of CCl_4 for Eastern China);
yes for the aspect I feel is most important (emission of CCl_4 from various industries).

Response:

For the aspect of the work of emission of CCl₄ for eastern China, we have changed to a more advanced method than in our first submission. In our revised manuscript, we have used inverse modeling and performed sufficient tests to support the conclusions and claims.

For the aspect of the work of emission of CCl₄ from various industries, thank you for the positive feedback. We have further improved the text. Please see the detailed responses below.

Are there any flaws in the data analysis, interpretation and conclusions? Do these prohibit publication or require revision?

Yes, as I will elaborate below. Simply, using the nation-wide estimate of CHCl₃ for two years, 2013 and 2015 (Supplemental Table 1) is flawed because the target gas, CCl₄, does not exhibit a strong enough correlation with CHCl₃ to justify the use of CHCl₃ as a tracer for the emission of CCl₄. Also, a lot has happened in China since 2015, following the 2018 publication of the Montzka et al. paper. Applying data from 2013 and 2015 to infer emissions in 2020 and 2021 is not sound. Finally and very importantly, the data could be used in an inverse model framework to compute emissions of CCl₄ from Eastern China. Yet this approach is not used.

Response:

Following the suggestions by reviewers, we have given up the interspecies correlation emission estimates and used emission inverse modeling in our revised manuscript. Thus, the flaws have been avoided, and the uncertainty value for this emission is justified (please see the detailed responses below).

Is the methodology sound? Does the work meet the expected standards in your field?

Methods not sound, as noted above.

In terms of expected standards in the field, there are a large number of problems including the reader being asked to believe there is no substantial bias between the standard used for Trinidad Head Station (the Scripps Institution of Oceanography USA25 standard) and the standard used at their station (the Linde Gas standard), with reference only in the literature and no support for supplement. In papers on this topic where mixed standards are used, it is 2 common to report side-by-side measurements of a sample from a current lab capable of running both standards, to quantify the actual bias).

The quantification of the emission of CCl₄ from Eastern China is well below the expected standard, which is to make use of an inverse (or perhaps data assimilation) model.

Response:

Following the suggestions by reviewers, in our revised manuscript (1), we have provided a citation of the inter-comparison between the SIO and Linde gas

calibration standards (the ratio of two standards is 1.02, which provides justifications to compare concentration data from stations using two standards), and (2) we have switched to use emission inverse modeling to avoid the flaws of interspecies correlation method mentioned above (please see the detailed responses below).

Is there enough detail provided in the methods for the work to be reproduced?

No. There is minimal discussion of either the Robust Extraction of Baseline Signals (REBS) or the trajectory clustering method in either Main, Methods, or Supplement.

Response:

In our revised manuscript, we have added detailed descriptions of either the robust extraction of baseline signals (REBS) in the main text or the trajectory clustering method in SI.

Now I will summarize the paper and highlight my “Major” and “Minor” points.

Overview:

First off, the paper is exceptionally well-written. Other than the occasional use of “This” or “These” as a noun, there is little I can offer to improve the communication of the message. I sincerely appreciate being sent a paper to review that is so well-written.

Response:

Thank you. As for our revised manuscript, we have also tried to improve the communication of messages.

The paper reports observations of CCl₄ and related species from two sites in Eastern China. The observations are stated to be hourly; 11,174 sample were acquired at one site between Jan 2021 and Dec 2022 whereas 294 samples were acquired at the other site between May 2021 and Nov 2022. For the site with the larger number of samples, that were collected for about 65% of the sampling opportunities (that is, 11,1174 / 2x8760). For the other site, data were collected for only about 2.5% of the sampling opportunities (this is, 294 / 16/12 x 8760)

The data are used to compared mean and various percentiles of CCl₄ to a time series from the Trinidad Head station, which reveals frequent enhancements of CCl₄ in Eastern China and to compute the emission of CCl₄ in Eastern China in 2020 and 2021. Finally, observations of CCl₄ are reported from the exhaust of 17 types of enterprises in Wuxi, which is in the Jiangsu province of Eastern China. These observations show highly elevated levels of CCl₄ in the following sections:

Manufacture of General Purpose Machinery (mainly manufacturing of engines, excavators, and other heavy machinery); Manufacture of Raw Chemical Materials and Chemical Products industry (mainly manufacturing coatings and resins), Manufacture of Electrical Machinery and Apparatus, and the Manufacture of Automobiles.

Response:

Thank you. Below, we have provided our point-by-point responses to each of the reviewers' comments.

In my opinion, the comparison of time series for CCl_4 to data from Trinidad Head (that is, Figure 1b) is likely not enough to warrant publication in Nature Communications, because the contemporary emission of CCl_4 from Eastern China is simply well known. The only possible surprise here would have been the disappearance of elevated CCl_4 from Eastern China. I state the elevation of CCl_4 in Eastern China is well known based on the content of numerous other studies cited in the paper and that have data which are used in Figure 1b, as well as a very important additional paper (Benish, S. E. et al., Airborne observations of CFCs over Hebei Province, China in Spring, JGR, 126, e2021JD035152, 2021) that is not cited (lots more on the importance of this paper below).

Response:

Thank you for the comment. The important additional reference of Benish et al. (2021) has been added in our revised manuscript and in our new Supplementary Fig. 3. We agree that based on previous studies, the elevation of CCl_4 concentrations and emissions in eastern China before 2019 is already well known; therefore, the presence of elevated CCl_4 is not surprising. However, it is surprising that after the 2019–2022 strict government supervision on CCl_4 , there were still elevated CCl_4 concentrations and emissions in eastern China during 2021–2022, as uncovered in this study. Following the suggestions, we have moved the concentration results and comparison with previous studies from the main text to SI.

In my opinion, the computation of the emission of CCl_4 for Eastern China in Nature Communications is also not warranted, because CCl_4 exhibits such a poor correlation with CHCl_3 , the gas selected as the “Tracer” for Equation 4. Figure 2 of the Supplement shows almost no correlation between these two species; the data look like a traditional “shotgun” blast, with the slope clearly influenced by a single outlier. Um, under what conditions was this outlier obtained? And, how does the slope change with the neglect of this single outlier? Any result that

depends on a single outlier is highly suspicious. Finally and most importantly, how can the slope of the data (“0.087” with an uncertainty of only “0.001”) be known so well, when the data (outside this one outlier) exhibit almost no correlation? Please note Benish et al. (their Figure 4) report a value of r^2 between CCl_4 and CHCl_3 of only 0.03 based on airborne sampling over the Hebei Province in spring of 2016. A much higher value of r^2 of 0.78 is reported between CCl_4 and CFC-11.

Response:

Thank you for this comment. We ultimately switched to using the inverse modeling method for emission estimation, and the above flaws of the interspecies correlation method have been avoided.

Finally, in my opinion, the most important part of this paper is not given due justice. I have extracted from the Potential industrial sectors of CCl_4 emissions section of the paper the four dominant industrial sectors. None of these sectors are named in the abstract, which only states:

Subsequently, our study identified potential industrial sources of CCl_4 emissions that were not revealed in previous studies and not controlled by the governmental supervision of CCl_4 . Therefore, our study can guide mitigations of CCl_4 emission and endorse continued monitoring of atmospheric CCl_4 to protect the ozone layer.

Response:

Thank you for the comment. In our revised manuscript, we have named the potential industrial sectors of CCl_4 emissions in the abstract.

Revised text

Lines 31–36 in the revised manuscript: “*Subsequently, our study identified potential industrial sources (manufacture of general purpose machinery and manufacture of raw chemical materials, and chemical products) of CCl_4 emissions that were not revealed in previous studies and were not under governmental supervision.*”

I shall provide further detail on a few Major and Minor points below.

Major Points

1) The numerical values of the emission of CCl_4 for Eastern China in 2021 and 2022 are highly suspicious, due to the use of CHCl_3 as “the tracer”, the use of emissions for

CHCl₃ from much earlier years (2013 & 2015), and the report by others (that is, Benish et al. 2021) of much higher correlations between CCl₄ and other species, such as CFC-11. If the authors see a high correlation of CCl₄ with CFC-11 that is not dependent on a single outlier, then perhaps CFC-11 can be used as “the tracer”. Regardless, whatever tracer, if the authors insist on following this approach, upon revision I suggest they:

- a) describe the correlation of the tracer with CCl₄
- b) state how the uncertainty in the slope is found
- c) justify the use of much older emission estimates (for CHCl₃, for 2013 and 2015) to a more recent time period, given so much attention to the emission of ODSs in China following the publication in 2018 of the Montzka et al. paper.
- d) Most importantly show that the selected tracer exhibits a meaningful correlation with CCl₄, not only for the ~2 year station data, but also for the samples obtained in the exhaust of 17 specific industries. If CHCl₃ show a tight correlation for data from Manufacture of General Purpose Machinery, Raw Chemical Materials and Chemical Products, Electrical Machinery and Apparatus, and Automobiles, then perhaps this gas can be used as a suitable tracer. The submitted paper does not describe any other data for the 17 specific industries, other than CCl₄. If there is little to no correlation between CHCl₃ and CCl₄, then another tracer is needed. Regardless, the selected tracer *MUST* show a significant correlation with CCl₄ for both the station data and the exhaust data, for the suspicion noted above to be assuaged.

Another option I encourage the authors to consider, should they insist on attempting to quantify the emission of CCl₄ for Eastern China in 2021 and 2022, is to use either the inverse model approach or perhaps data assimilation.

One more option, one I prefer, is that if this paper proceeds in Nature Communications, the finding of significant emissions of CCl₄ from certain industrial sectors be the primary point of emphasis. I expand further on this suggestion in Major Point #4, below.

Response:

We agree that the interspecies correlation method used to estimate CCl₄ emissions in our previous manuscript has several flaws, as pointed out in this comment. Thus, following the suggestions, we have abandoned using the interspecies correlation method and instead used the inverse modeling method in our revised manuscript. The inverse modeling can directly quantify total CCl₄ emissions in eastern China, avoiding the interspecies correlation flaws of (1) good correlations between CCl₄ and tracer and (2) known emissions of tracer during 2021–2022. The inverse modeling method can

also provide spatial distributions ($1^\circ \times 1^\circ$ in this study) of CCl_4 , while the interspecies correlation cannot.

As suggested by this comment, the findings of significant emissions of CCl_4 from specific industrial sectors were deepened in our revised manuscript. The specific changes in the article are listed in Major Point #4.

2) I suggest the authors provide better justification of the combination of data from Trinidad Head Station obtained using the Scripps Institution of Oceanography USA25 standard with data from their stations, that use the Linde Gas standard. Seeing a comparison of samples run through a lab with one standard, against data measured using the other standard, is common (for example, Tables S2, S3, and S4 of the previously mentioned Benish et al. paper). Also, upon revision, there are many places where the results of Benish et al. 2021 should be cited, and data from this paper should be added to Figure 2b.

Response:

Thank you for this comment. Benish et al. (2021) already did the test on CCl_4 concentrations between Scripps Institution of Oceanography and Linde Gas standards. They found that the ratio of CCl_4 concentrations quantified using the Scripps Institution of Oceanography standard to those using the Linde Gas standard was 1.02, suggesting the difference between the CCl_4 concentrations quantified based on these two standard gases is negligible (2%). Therefore, no additional tests were performed in this study. Moreover, concentration data from Trinidad Head Station (and AGAGE stations of Mace Head, Jungfraujoch, and Ragged Point are added in our revised manuscript) using the Scripps Institution of Oceanography standard are only introduced for comparison with our measured concentrations at ZJU and SHH in Supplementary Fig. 3. Moreover, the CCl_4 concentrations measured by Benish et al.¹ has been added to original Figure 2b (now Supplementary Fig.3) in our revised manuscript.

Table S4 in Benish et al.. Summary statistics (N=27) and average CMA/PKU correction factor for 12 halocarbons (pptv) quantified during ARIAs. The first number denotes the uncorrected value and the second number shows the corrected concentration using the average CMA/PKU ratio obtained from the intercomparison

experiment.

Compound	Mean	STD	Min	25 th	50 th	75 th	Max	CMA/PKU Correction
Carbon tetrachloride	88/89	21/21	53/54	74/75	86/88	102/104	138/140	1.02

Revised text:

Lines 15–19 in the revised Supplementary Information: “The four AGAGE stations used SIO-05 standard gas (Scripps Institution of Oceanography, USA²) calibration scale, while this study used Linde Gas, thus introducing a potential bias. Nevertheless, the differences between these standard gases are likely small^{1,3,4}, e.g., the ratio between CCl₄ concentrations determined with SIO-05 standard gas and Linde Gas was 1.02¹”.

Revised figure:

Supplementary Fig. 3 Concentration levels of CCl₄ in different studies.

Uncertainties are represented with error bars. The concentrations of CCl₄ at ZJU and SHH are shown with box and whisker plots, the 5th, 25th, 75th, 95th percentiles and

mean levels are shown.

Reference:

1 Benish, S. E., Salawitch, R. J., Ren, X., He, H. & Dickerson, R. R. Airborne Observations of CFCs Over Hebei Province, China in Spring 2016. *J. Geophys. Res-Atmos.* 126 (2021).

2 Prinn, R. G. et al. History of chemically and radiatively important atmospheric gases from the Advanced Global Atmospheric Gases Experiment (AGAGE). *Earth Syst. Sci. Data* 10, 985-1018 (2018).

3 Hu, L. et al. Continued emissions of carbon tetrachloride from the United States nearly two decades after its phaseout for dispersive uses. *Proc. Natl. Acad. Sci. U. S. A.* 113, 2880-2885 (2016).

4 Fang, X. et al. Ambient mixing ratios of chlorofluorocarbons, hydrochlorofluorocarbons and hydrofluorocarbons in 46 Chinese cities. *Atmos. Environ.* 54, 387-392 (2012).

3) As noted above, there is minimal description of either the Robust Extraction of Baseline Signals (REBS) or the trajectory clustering method in either Main, Methods, or Supplement. Upon revision, more details are needed. For the trajectory clustering method, the HYSPLIT model is used, yet there are no references for HYSPLIT apparent in the submitted manuscript. I suggest citing this paper by Stein et al. <https://doi.org/10.1175/BAMS-D-14-00110.1>. Also, Figure 1 of Supplement is hard to understand. It is not clear which colors correspond to which numbered trajectory, and the use of yellow (for SHH) is illadvised, as this line is very hard to see. I also suggest writing “ZJU” onto both panels of Fig 1a and “SHH” on both panels of Fig 1b.

Response:

Thank you for this comment. Our revised manuscript has added a more detailed description of REBS and the trajectory clustering method. Reference to Stein et al. (2015) has been cited. And we have redrawn the original Supplementary Fig.1 (now Supplementary Fig. 6).

Revised text:

For REBS method, more detailed introduction has added in the revised manuscript in lines 295-308: “The robust extraction of baseline signal (REBS) method⁵ was applied to distinguish background and non-background concentrations of

the in-situ $C\text{Cl}_4$ concentrations at ZJU. The REBS is a statistical method developed by Ruckstuhl et al.⁵ to extract background signals using a robust local regression model, and has been widely applied to determine baselines of trace gases in inversion studies^{6,7}. The observed concentration at a certain time ($y(t_i)$) was divided into three parts as shown depicted in the following equation:

$$y(t_i) = g(t_i) + m(t_i) + e_i$$

where $g(t_i)$ represents the background concentration at time t_i , $m(t_i)$ is the enhanced concentration caused by polluted plum during t_i , and e_i represents the observational error.

The baseline curve g was obtained using the REBS technique over a sufficiently long bandwidth (90 days) by assuming that most observations are at background levels and that the baseline signal changes slowly relative to the regional signal. In this method, data points closer to the time of consideration were given more weight, and data points outside a specific range (1.5σ in this study) were iteratively excluded.”

For HYSPLIT model, the reference related to HYSPLIT was added in the revised Supplementary Information in line 51-53, “back trajectories of the sampled air masses were analyzed using the Hybrid Single-Particle Lagrangian Integrated Trajectory Model (HYSPLIT) (<http://www.arl.noaa.gov/ready/hysplit4.html>)⁸”.

Cluster method of air mass was added in lines 59-65: “Then, backward trajectories gained from HYSPLIT were analyzed based on the Euclidean distance clustering algorithm using the Geographical Information System (GIS)-based TrajStat software (<http://meteothink.org/docs/trajstat/index.html>) developed by Wang et al.⁹ to distinguish different types of air mass. The equation for calculating Euclidean distance between trajectories is as follows¹⁰:

$$d_{12} = \sqrt{\sum_{i=1}^n ((X_1(i) - X_2(i))^2 + (Y_1(i) - Y_2(i))^2)} \quad (1)$$

where $X_1(Y_1)$ and $X_2(Y_2)$ represent the backward trajectories 1 and 2, respectively.”

Revised figure:

Supplementary Fig.6 Clusters of backward trajectories during the sampling period and corresponding CCl₄ concentration. a Cluster analysis of 120 h backward trajectories for ZJU during the sampling period calculated using the HYSPLIT model, with the starting height at 100 m above ground level. Running intervals were set as 1 h for each day; the ratio, moving height, and average concentrations of CCl₄ of each cluster are also presented. **b** Same as (a), excluding the SHH station.

Reference:

5 Ruckstuhl, A. F. et al. Robust extraction of baseline signal of atmospheric trace species using local regression. Atmospheric Measurement Techniques 5, 2613-2624 (2012).

- 6 Affolter, S. et al. Assessing local CO₂ contamination revealed by two near-by high altitude records at Jungfrauoch, Switzerland. *Environ. Res. Lett.* 16 (2021).
- 7 Western, L. M. et al. A renewed rise in global HCFC-141b emissions between 2017–2021. *Atmos. Chem. Phys.* 22, 9601-9616 (2022).
- 8 Stein, A. F. et al. NOAA's HYSPLIT Atmospheric Transport and Dispersion Modeling System. *Bulletin of the American Meteorological Society* 96, 2059-2077 (2015).
- 9 Wang, Y. Q., Zhang, X. Y. & Draxler, R. R. TrajStat: GIS-based software that uses various trajectory statistical analysis methods to identify potential sources from long-term air pollution measurement data. *Environ. Model. Software* 24, 938-939 (2009).
- 10 Sirois, A. & Bottenheim, J. W. Use of backward trajectories to interpret the 5-year record of PAN and O₃ ambient air concentrations at Kejimikujik National Park, Nova Scotia. *J. Geophys. Res-Atmos.* 100, 2867-2881 (1995).

4) As noted above, I feel the most important result of this paper is not highlighted adequately. If this paper proceeds in Nature Communication, I suggest dispensing with the comparison of the new measurements of CCl₄ in 2020 and 2021 with data from Trinidad Head quickly, either improving or perhaps striking the attempt to compute emissions of CCl₄ for Eastern China over 2020 and 2021, then moving a lot of the material in Supplement regarding what I believe is the key new message of this paper into Main (or Methods).

Response:

Thank you for this comment. Following the suggestions, (1) more discussion on industrial sectors investigated in this study has been added; (2) comparison of the new measurements of CCl₄ in 2020 and 2021 with data from Trinidad Head has been moved to Supplementary Information; (3) a lot of the material in SI has been moved into Main; (4) emissions of CCl₄ for eastern China over 2021 and 2022 was recalculated with inverse model method rather than the interspecies correlation method previously used.

For industrial sector:

Revised text:

Discussion on implications from these newly discovered potential sources of CCl₄ has been added in lines 230-242 in the revised manuscript: *“Based on the results of this study, high concentrations of CCl₄ in 17 industries were observed in enterprises with low annual revenue (Supplementary Fig. 7). Specifically, the CCl₄ concentration was negatively correlated with the logarithm of the enterprise revenue ($R^2 = 0.49$, $p < 0.01$). Similar relationships between CCl₄ concentration and enterprise revenue were also observed in the MGPM ($R^2 = 0.75$, $p < 0.01$) and MRCM ($R^2 = 0.49$, $p < 0.01$) industries.*

The number of enterprises with annual revenue > 20 million CNY has exhibited an obvious upward trend for the MGPM and MEMA industries since 2018 (growth rate: 10–16% yr⁻¹ during 2018–2021)¹¹ (Supplementary Fig. 8). Although the number of enterprises in most other industries was relatively stable during 2011– 2021¹² (Supplementary Fig. 8), the overall trend has been on the rise (Supplementary Fig. 8). Considering that they may represent potential sources of CCl₄ emissions, attention should be paid to these industries (particularly MGPM) to evaluate their impact on the ozone layer.

This study identified previously unreported potential industrial sources of CCl₄ emissions. These findings can guide mitigation strategies and supervision of CCl₄ emissions, while strongly endorsing continued monitoring of atmospheric CCl₄ to protect the ozone layer.”

Revised figure:

Supplementary Fig. 7 Relationship between CCl₄ concentrations and revenue. a CCl₄ concentrations across revenue ranges for all sampled enterprises. **b.** Same as **a**, excluding MGPM. **c.** Same as **a**, excluding MRCM. The size of the dots represents the number of enterprises.

Supplementary Fig. 8 Number of enterprises above designated size (annual revenue > 20 million CNY) for each industry¹².

Reference

11 National Bureau of Statistics (NBS). China Statistical Yearbook. (China Statistics Press, 2021).

12 National Bureau of Statistics (NBS). China Statistical Yearbook. (China Statistics Press, 2011-2021).

For inverse modeling of CCl₄ emission in eastern China:

Revised text:

Description on the inverse modelling method has been added in lines 269-338 in the revised manuscript: “This study used an inverse modeling technique based on a FLEXPART atmospheric transport simulation model and the Bayesian inversion algorithm to quantify CCl₄ emissions. The FLEXPART-based inversion method has

been applied in many previous studies¹³⁻¹⁵. In brief, driven by the meteorological data (European Centre for Medium-Range Weather Forecasts) with a spatial resolution of $1^\circ \times 1^\circ$ and a temporal resolution of 3 h, the FLEXPART model was ran in backward mode for 20 days. The source–receptor relationship (termed as “emission sensitivity”) matrix was established based on the backward simulation. Combining the derived emission sensitivity matrix, Bayesian inversion technique, and the CCL_4 data from ZJU and SHH, yielded the CCL_4 emission strength ($1^\circ \times 1^\circ$) in grid cells over eastern China. The associated equations are as follows:

$$J(x) = \frac{1}{2} (x - x_a)^T S_a^{-1} (x - x_a) + \frac{1}{2} (y^{\text{obs}} - Hx)^T S_o^{-1} (y^{\text{obs}} - Hx)$$

By solving $\nabla_x J(x) = 0$ yields:

$$x = x_a + S_a H^T (H S_a H^T + S_o)^{-1} (y^{\text{obs}} - Hx_a)$$

$$S_b = (H^T S_o^{-1} H + S_a^{-1})^{-1}$$

where x represents the state vector of the emission strength, y^{obs} represents the CCL_4 measurement vector, x_a represents the prior emission vector, H is the emission sensitivity, S_a and S_b are the error covariance matrix of prior and posterior emissions, respectively, and S_o represents the error covariance matrix of measurement data. To obtain the prior emission vector (x_a), the national total CCL_4 emissions in China during 2011–2015 ($23.6 \pm 7.1 \text{ Gg yr}^{-1}$)¹⁶ and emissions in other countries (derived from global emissions [$44 \pm 15 \text{ Gg yr}^{-1}$ in 2020]¹⁷ and emission estimates of the United States [$4.0 (2.0\text{--}6.5) \text{ Gg yr}^{-1}$ during 2008-2012]³, Japan [0.6 Gg yr^{-1} in 2014]¹⁸, South Korea [0.2 Gg yr^{-1} in 2014]¹⁸, and India [2.8 Gg yr^{-1} in 2014]¹⁸) were assigned to grid cells with a uniform spatial distribution in eastern China, and a population-proxy distribution in other regions (Supplementary Fig. 9a). The inversion results were also evaluated using prior emissions following population distribution in 2020¹⁹ (Supplementary Fig. 9b).

The robust extraction of baseline signal (REBS) method⁵ was applied to distinguish background and non-background concentrations of the in-situ CCL_4 concentrations at ZJU. The REBS is a statistical method developed by Ruckstuhl et al.⁵ to extract background signals using a robust local regression model, and has been widely applied

to determine baselines of trace gases in inversion studies^{6,7}. The observed concentration at a certain time ($y(t_i)$) was divided into three parts as shown depicted in the following equation:

$$y(t_i) = g(t_i) + m(t_i) + e_i$$

where $g(t_i)$ represents the background concentration at time t_i , $m(t_i)$ is the enhanced concentration caused by polluted plum during t_i , and e_i represents the observational error.

The baseline curve g was obtained using the REBS technique over a sufficiently long bandwidth (90 days) by assuming that most observations are at background levels and that the baseline signal changes slowly relative to the regional signal. In this method, data points closer to the time of consideration were given more weight, and data points outside a specific range (1.5σ in this study) were iteratively excluded.

For flask samples at SHH, the background concentration of CCL_4 was determined as the lowest concentration measured in a two-month moving window. The observational error (σ_{obs}) was calculated as follows:

$$\sigma_{obs} = \sqrt{\sigma_{obs_precision}^2 + \sigma_{obs_representation}^2 + \sigma_{background}^2}$$

where $\sigma_{obs_precision}$ is the measurement precision of CCL_4 , $\sigma_{obs_representation}$ stands for representation of the observation, $\sigma_{background}$ represents the background uncertainty. In this study, $\sigma_{obs_representation}$ was calculated as 1-Sigma standard deviation of the measurements each day for online samples and as 1-Sigma standard deviation of the measurements in a two-month moving window for flask samples. The $\sigma_{background}$ was estimated as 1-Sigma standard deviation of the fitted background concentrations during the sampling period. The diagonal elements of S_o were set to squared σ_{obs} .

The diagonal elements of S_a were calculated as squared uncertainty of the prior emission field. The off-diagonal elements of S_a were calculated based on previous studies^{15,20}, with the spatial decorrelation length scale set to 400 Km. In this study, three sets of prior emissions were established (i.e., 150%, 100% and 50% of the reference

prior emissions), and three uncertainties (600%, 450%, 300% for uniformly distributed prior emissions) were set under each set of prior emissions. A total of nine inversions were carried out for each year. The final posterior emissions were the average of the nine inversions (Supplementary Fig. 10). The posterior CCl₄ emissions in eastern China under the nine inversions in 2021 and 2022 were loaded in the range of the three sets of prior emissions (Supplementary Fig. 10). The posterior emissions were insensitive (varied < 9%) to variations in emission uncertainty, suggesting that the prior emissions used in this study were not systematically high or low, and that the observations constrain posterior emissions well. We also tested baselines from the AGAGE baseline filtering method (baseline obtained by fitting the daily minimum with a second-order polynomial)²¹ to examine the impact of different baselines on posterior emissions. The inversion ensembles (Supplementary Table 3) using two baselines (AGAGE and REBS methods) and two prior emission fields (uniform and population-proxy distributions) show that the posterior emissions varied by < 6% (standard deviation divided by mean) during 2021–2022 (Supplementary Fig. 11).”

The inversed emission of CCl₄ in eastern China was described in lines 122-141 in the revised manuscript: “Using the atmospheric concentration observations at ZJU and SHH and an inverse modeling method, CCl₄ emissions in eastern China were quantified (see Methods for detailed information). Eastern China in this study includes Shandong, Henan, Shanghai, Jiangsu, Zhejiang, Anhui, Jiangxi, and Fujian, the emissions from which were detected in the mole fractions observed at ZJU and SHH sites (Fig. 1). CCl₄ emissions in eastern China were estimated to be 7.0 ± 2.7 Gg yr⁻¹ in 2021, and 8.2 ± 3.3 Gg yr⁻¹ in 2022, with a mean emission of 7.6 ± 3.1 Gg yr⁻¹ (Fig. 2b). Considering that previous top-down studies reported CCl₄ emissions until 2019, this study provides the first top-down estimates for the 2021–2022 period.

Park et al.²² reported that CCl₄ emissions in eastern China (see definition of boundary in Park et al.²²) decreased from 11.9 ± 1.5 Gg yr⁻¹ in 2014 to 8.8 ± 1.7 Gg yr⁻¹ in 2017, 6.0 ± 1.5 Gg yr⁻¹ in 2018, and 6.3 ± 1.0 Gg yr⁻¹ in 2019 (Fig. 2b). Meanwhile, the CCl₄ emissions in eastern China (7.6 ± 3.1 Gg yr⁻¹) estimated in this study for 2021–2022 were comparable to (higher than, although not significantly) those in 2019 ($6.3 \pm$

1.0 Gg yr⁻¹)²² within the uncertainties, suggesting persistent CCl₄ emissions in these areas since 2019. Additionally, the CCl₄ emissions in eastern China reported in this study (7.6 ± 3.1 Gg yr⁻¹ in 2021–2022) and by Park et al.²² are lower than those for all of China reported by Park et al.¹⁶ (23.6 ± 7.1 Gg yr⁻¹ during 2011–2015) and Lunt et al.²³ (17 (11–24) Gg yr⁻¹ during 2009–2016) (Fig. 2b), indicating the presence of CCl₄ emissions in non-eastern regions of China.”

The spatial distribution of CCl₄ emissions in eastern China based on the inversed results was shown in lines 151-155 in the revised manuscript: *“The northeastern region of Shandong, eastern region of Jiangsu, and southeastern region of Zhejiang were identified in this study as major CCl₄ source regions in eastern China during 2021–2022 (Fig. 3, Supplementary Fig. 4 and Supplementary Fig. 5). Additionally, air mass corresponding to high CCl₄ concentrations typically pass through these regions (Supplementary Fig. 6, see Supplementary Discussion 2 for more details).”*

Revised figure:

Fig.2 Concentration levels and emission estimates of CCl_4 . **a** Time series of CCl_4 concentrations at ZJU and SHH during 2021–2022. **b** Comparison of CCl_4 emission estimates in this study with those from previous studies.

Fig. 1 Mean Spatial distribution of CCl_4 emissions in eastern China derived from uniformly distributed prior emissions. a For 2021. b For 2022. Spatial distributions of CCl_4 emissions from each inversion using REBS or AGAGE baseline filtering method are shown in Supplementary Fig. 4. Supplementary Fig. 5 is the same as Supplementary Fig. 4 using population-proxy prior emissions.

Supplementary Fig. 4 Spatial distribution of CCl₄ emissions in eastern China derived from uniformly distributed prior emissions. a, b Based on baseline extracted with REBS method. **c, d** same as a, b, but the baseline is extracted with the method used by AGAGE. The blue triangle and dot represent the ZJU and SHH sites, respectively.

Supplementary Fig. 2 Spatial distribution of CCl_4 emissions in eastern China derived from population-proxy distributed prior emissions. a, b Based on baseline extracted with REBS method. **c, d** same as a, b, but the baseline is extracted with the method used by AGAGE. The blue triangle and dot represent the ZJU and SHH sites, respectively.

Supplementary Fig. 9 CCl₄ prior emissions used in this study. a Prior emissions in eastern China set to a uniform distribution. **b** Prior emissions in eastern China distributed based on population density. The blue triangle and dot represent the ZJU and SHH sites, respectively.

Supplementary Fig. 10 Posterior CCl_4 emissions in eastern China. *a* Nine inversions (three magnitudes of 150%, 100%, and 50% of reference prior emission multiplied by the three magnitudes of 600%, 450%, and 300% of reference prior emission uncertainty) for CCl_4 emissions in eastern China in 2021 and 2022 based on the baseline extracted with the REBS method and uniformly distributed prior emissions (Inv-REBS-UniformPrior). *b* Same as *a* but based on the baseline extracted with the method suggested by AGAGE (Inv-AGAGE-UniformPrior). *c* Same as *a* but based on the non-uniformly distributed prior emissions (Inv-REBS-PopPrior). *d* Same as *c* but based on the baseline extracted with the method suggested by AGAGE (Inv-AGAGE-PopPrior). The three green horizontal solid lines represent three magnitudes of 150%, 100%, and 50% of reference prior emissions for eastern China.

Supplementary Fig. 11 *CCl₄ emissions from four inversion sets for eastern China.* Detailed information on four inversion sets using two baselines (AGAGE and REBS methods) and two prior emission fields (uniform and population-proxy distributions) is shown in Supplementary Table 3.

Supplementary Table 3 *The set-up for the four inversion frameworks*

Category	Baseline extraction method	Priori emission distribution
Case A	REBS	Uniformly distributed in eastern China, and distributed based on population density in other areas
Case B	AGAGE	Uniformly distributed in eastern China, and distributed based on population density in other areas
Case C	REBS	Distributed based on population density
Case D	AGAGE	Distributed based on population density

Reference:

3 Hu, L. et al. Continued emissions of carbon tetrachloride from the United States nearly two decades after its phaseout for dispersive uses. *Proc. Natl. Acad. Sci. U. S. A.* 113, 2880-2885 (2016).

5 Ruckstuhl, A. F. et al. Robust extraction of baseline signal of atmospheric trace species using local regression. *Atmospheric Measurement Techniques* 5, 2613-2624 (2012).

6 Affolter, S. et al. Assessing local CO₂ contamination revealed by two near-by

- high altitude records at Jungfrauoch, Switzerland. *Environ. Res. Lett.* 16 (2021).
- 7 Western, L. M. et al. A renewed rise in global HCFC-141b emissions between 2017–2021. *Atmos. Chem. Phys.* 22, 9601-9616 (2022).
- 13 Fang, X. et al. Rapid increase in ozone-depleting chloroform emissions from China. *Nature Geosci.* 12, 89-93 (2019).
- 14 Fang, X. et al. Changes in HCFC Emissions in China During 2011–2017. *Geophys. Res. Lett.* 46, 10034-10042 (2019).
- 15 Fang, X. et al. Multiannual top-down estimate of HFC-23 emissions in East Asia. *Environ. Sci. Technol.* 49, 4345-4353 (2015).
- 16 Park, S. et al. Toward resolving the budget discrepancy of ozone-depleting carbon tetrachloride (CCl₄): an analysis of top-down emissions from China. *Atmos. Chem. Phys.* 18, 11729-11738 (2018).
- 17 World Meteorological Organization (WMO). *Scientific Assessment of Ozone Depletion: 2022*, GAW Report No. 278. 509 (World Meteorological Organization (WMO), Geneva, 2022).
- 18 Sherry, D., McCulloch, A., Liang, Q., Reimann, S. & Newman, P. A. Current sources of carbon tetrachloride (CCl₄) in our atmosphere. *Environ. Res. Lett.* 13, 024004 (2018).
- 19 Center for International Earth Science Information Network - CIESIN - Columbia University. (NASA Socioeconomic Data and Applications Center (SEDAC), Palisades, New York, 2018).
- 20 Brunner, D. et al. An extended Kalman-filter for regional scale inverse emission estimation. *Atmos. Chem. Phys.* 12, 3455-3478 (2012).
- 21 S. O'Doherty et al. In situ chloroform measurements at Advanced Global Atmospheric Gases Experiment atmospheric research stations from 1994 to 1998. *J. Geophys. Res.* 106, 20429-20444 (2001).
- 22 Park, S. et al. A decline in emissions of CFC-11 and related chemicals from eastern China. *Nature* 590, 433-437 (2021).
- 23 Lunt, M. F. et al. Continued Emissions of the Ozone-Depleting Substance Carbon Tetrachloride From Eastern Asia. *Geophys. Res. Lett.* 45, 11423-11430 (2018).

Minor points

a) The ODP and GWP for CCl₄ given in the most recent Scientific Assessment Ozone Depletion report are 0.87 and 2150, respectively. Upon revision, please adopt these values (since they are current and have been assessed by the relevant experts) on lines 38 and 39, and in the rest of the paper (that is, CFC-11 equivalent emissions) that rely on the use of these numbers.

Response:

Thank you for this comment. In our revised manuscript, the ODP and GWP values of 0.87 and 2150 have been used in the original lines 37 and 38 and in the rest of the paper (CFC-11 equivalent emissions).

Revised text:

The revised ODP and GWP value was shown in lines 36-38 in the revised

manuscript: “Carbon tetrachloride (CCl₄) is a first-generation ozone-depleting substance (ODS) with an ozone depletion potential of 0.87 and a potent greenhouse gas with a global warming potential of 2150¹⁷”.

The recalculated ODP and GWP value was shown in lines 175-187 in the revised

manuscript: “For instance, the ozone depletion potential (ODP)-weighted emissions (equivalent to CFC-11 emissions) of CCl₄ in eastern China estimated in this study (6.6 ± 2.7 CFC-11-eq Gg yr⁻¹) are comparable to the unexpected emissions of CFC-11 (7.0 ± 4.0 Gg yr⁻¹) in eastern China during the 2014–2017 period relative to those during 2008–2012²²..... Moreover, ODP-weighted CCl₄ emissions in eastern China (6.6 ± 2.7 CFC-11-eq Gg yr⁻¹) are several times higher than the ODP-weighted emissions of other major ODSs, for example, it is 17.6 times higher than the 0.38 ± 0.75 CFC-11-eq Gg yr⁻¹ of dichlorodifluoromethane (CFC-12) in 2019 in eastern China²², 8.4 times the 0.79 (0.65 – 1.1) CFC-11-eq Gg yr⁻¹ of 1,1-dichloro-1-fluoroethane (HCFC-141b) in 2020 in eastern China⁷, and 1.4 times the 4.6 (3.9 – 5.3) CFC-11-eq Gg yr⁻¹ of chlorodifluoromethane (HCFC-22) in 2019 (but for all of China²⁴).”

Reference:

⁷ Western, L. M. et al. A renewed rise in global HCFC-141b emissions between 2017–2021. *Atmos. Chem. Phys.* 22, 9601-9616 (2022).

¹⁷ World Meteorological Organization (WMO). Scientific Assessment of Ozone Depletion: 2022, GAW Report No. 278. 509 (World Meteorological Organization

(WMO), Geneva, 2022).

22 Park, S. et al. A decline in emissions of CFC-11 and related chemicals from eastern China. *Nature* 590, 433-437 (2021).

24 Wu, J., Li, T., Wang, J., Zhang, D. & Peng, L. Establishment of HCFC-22 National-Provincial-Gridded Emission Inventories in China and the Analysis of Emission Reduction Potential. *Environ. Sci. Technol.* 56, 814-822 (2022).

b) Line 50. Again, as noted above, the paper is extremely well written! Minor point, but rather than “to restrict CCl₄ emissions”, I suggest “to understand the reason for the gap in bottom-up and top-down emissions”. Regardless, if the paper is revised as suggested above, there will certainly be much new text regarding how the observations will be used in the future “to restrict emissions”.

Response:

Thank you for this comment. The “to restrict CCl₄ emissions” has been changed to “to understand the reason for the gap in bottom-up and top-down emissions” in our revised manuscript.

Revised text:

Lines 47-49 in the revised manuscript: “*Therefore, further CCl₄ atmospheric observations are needed to understand the reason for the gap in bottom-up and top-down emissions.*”

c) Benish et al., multiple lines: Benish et al. could also be cited on line 68 as well as lines 76 to 79. : Benish et al. also showed, based on their strong correlation of CFC-11 with CCl₄, the likely co-production of these two species (line 82).

Response:

Thank you for this comment. In our revised manuscript, the suggested reference of Benish et al. (2021) has been added.

Revised text:

Lines 74-77 in the revised manuscript : “*Although there are reported atmospheric observations of CCl₄ within eastern China, namely within Nanjing (120 ± 30 ppt; parts per trillion) in 2018²⁵, Dongying (129 ± 62 ppt) in 2017²⁶, Hebei (89 ± 21 ppt) in 2016¹, and Lin’an (111 ± 11 ppt) in 2001²⁷”.*

Line 81-83 in the revised manuscript: “Based on the strong correlation between CFC-11 and CCl₄ ($R^2 = 0.78$), Benish et al.¹ showed the likely co-production of these two species.”

Reference:

1 Benish, S. E., Salawitch, R. J., Ren, X., He, H. & Dickerson, R. R. Airborne Observations of CFCs Over Hebei Province, China in Spring 2016. *J. Geophys. Res.-Atmos.* 126 (2021).

25 Fan, M. Y. et al. Source apportionments of atmospheric volatile organic compounds in Nanjing, China during high ozone pollution season. *Chemosphere* 263, 128025 (2021).

26 Zheng, P. et al. Characteristics and sources of halogenated hydrocarbons in the Yellow River Delta region, northern China. *Atmos. Res.* 225, 70-80 (2019).

27 Wang, T. Relationships of trace gases and aerosols and the emission characteristics at Lin'an, a rural site in eastern China, during spring 2001. *J. Geophys. Res.* 109, D19S05 (2004).

d) Line 81. “observed” should be “reported”, since the paper was published in 1981.

Response:

Thank you for this comment. In our revised manuscript, “observed” has been replaced by “reported.” The paper was published in 2018 (Montzka, S. A. et al. An unexpected and persistent increase in global emissions of ozone-depleting CFC-11. *Nature*, 2018, 557, 413-417).

Revised text:

Line 81 in the revised manuscript: “In 2018, an unexpected rise in global trichlorofluoromethane (CFC-11) emission was reported.”

e) Lines 84, 86, and 88. The use of “16” on lines 84 & 86 and “19” on line 88 is confusing. If these numbers are truly different, a bit more explanation would be helpful.

Response:

Thank you for the reminder. The number of CCl₄ by-product enterprises under on-site

supervision was varied with time, it is 16 in 2019 and 19 in 2021. The sentence has been revised as follows:

Revised text:

Lines 89-91 in the revised manuscript: “By 2021, the number of CCl₄ by-product enterprises under on-site supervision had increased to 19²⁸, comprising verifiable and quantifiable CCl₄ online production monitoring systems connected to the central government monitoring platform²⁹.”

Reference:

28 Ministry of Ecology and Environment of China. Progress of ecological and environmental law enforcement work and next work arrangements (in Chinese), <https://baijiahao.baidu.com/s?id=1698552840313072252&wfr=spider&for=pc> (2021).

29 Ministry of Ecology and Environment of China. Record of the regular press conference of the Ministry of Ecology and Environment in April (in Chinese), https://www.mee.gov.cn/xxgk2018/xxgk/xxgk15/202104/t20210428_831177.html (2021).

f) Line 95. Suggest “this study describes observations of ...”

Response:

Thank you for this comment. We have revised the text following the suggestion.

Revised text:

Lines 98-99 in the revised manuscript: “Consequently, this study describes observations of atmospheric CCl₄ concentrations at two sites in eastern China...”

g) Line 99. Here, I prefer “substantial emissions of CCl₄” since substantial modifies emissions

Response:

Thank you for this comment. In our revised manuscript, we changed following the suggestion.

Revised text:

Lines 101-103 in the revised manuscript: *“This study reports substantial emissions of in eastern China after the dispersive use of CCl₄ was phased out in 2010.”*

h) Line 121. Not sure “41N” is really “close” to 30.306 and 28.583N; perhaps a different word can be used

Response:

Thank you for this comment. In our revised manuscript, we changed following the suggestion.

Revised text:

Lines 5-10 in the revised Supplementary Information: *“The concentrations were higher than the measurement in the four remote northern hemispheric stations (Mace Head (75.9 ± 0.23 ppt; 53.327°N , 9.904°W), Jungfrauoch (75.4 ± 0.29 ppt; 46.548°N , 7.985°E), Trinidad Head (76.0 ± 0.22 ppt; 41.054°N , 124.151°W), and Ragged Point (75.9 ± 0.26 ppt; 13.165°N , 59.432°W)) from the Advanced Global Atmospheric Gases Experiment (AGAGE; <http://agage.mit.edu/>) in 2021.”*

i) Line 125. Suggest “These observations indicate ...”

Response:

Thank you for this comment. In our revised manuscript, we changed following the suggestion.

Revised text:

Lines 11-12 in the revised Supplementary Information: *“These observations indicate the presence of substantial CCl₄ emissions in eastern China...”*

j) Line 131. In addition to the request for more information about REBS, the inclusion of a reference here would be helpful

Response:

Thank you for this comment. In our revised manuscript, we have changed following the suggestion. The reference to Ruckstuhl et al.⁵ has been added, and more detailed information has been presented in the method section (please see the response to major point 3).

Reference:

5 Ruckstuhl, A. F. et al. Robust extraction of baseline signal of atmospheric trace species using local regression. *Atmospheric Measurement Techniques* 5, 2613-2624 (2012).

k) Line 135. Suggest “respectively” rather than “in respect”

Response:

Thank you for this comment. In our revised manuscript, we have changed following the suggestion.

Revised text:

Lines 20-22 in the revised Supplementary Information: “*approximately 30% of the CCl₄ concentrations at ZJU (averaged as 93.9 ± 14.4 ppt) were identified as non-background concentrations, respectively.*”

l) Lines 146 to 149. Not sure all of this detail is needed, especially if the paper’s focus is altered, but if this text survives, should also cite numerical values from the Benish et al. study

Response:

Thank you for this comment. We believe it is necessary to show the enhanced concentration of CCl₄ to indicate the possible presence of ongoing CCl₄ emissions after 2010. Following your suggestion, our revised manuscript has added CCl₄ concentrations reported by Benish et al.

Revised text:

Lines 28-30 in the revised Supplementary Information: “*...e.g., enhanced CCl₄ concentrations in Beijing (approximately 34 ppt in 2015)³⁰, Nanjing (approximately 32 ppt in 2018)²⁵, Lushan (approximately 94 ppt during 2011–2012)³¹, Hebei (approximately 10 ppt in 2016)^l...*”

Reference:

1 Benish, S. E., Salawitch, R. J., Ren, X., He, H. & Dickerson, R. R. Airborne Observations of CFCs Over Hebei Province, China in Spring 2016. *J. Geophys. Res.-Atmos.* 126 (2021).

25 Fan, M. Y. et al. Source apportionments of atmospheric volatile organic compounds in Nanjing, China during high ozone pollution season. *Chemosphere* 263, 128025 (2021).

30 Li, J. et al. Effects of rigorous emission controls on reducing ambient volatile organic compounds in Beijing, China. *Sci. Total Environ.* 557-558, 531-541 (2016).

31 Yang, M., Wang, Y., Chen, J., Li, H. & Li, Y. Aromatic Hydrocarbons and Halocarbons at a Mountaintop in Southern China. *Aerosol Air Qual. Res.* 16, 478-491 (2016).

m) Lines 153 to 156. The use of HYSPLIT for the trajectory clustering should be stated, with a reference to a HYSPLIT paper, perhaps the above mentioned Stein et al. study

Response:

Thank you for this comment. Following the suggestion, the use of HYSPLIT for the trajectory clustering has been stated, and the suggested reference of Stein et al. has been added:

Revised text:

Lines 37-38 in the revised Supplementary Information: “Using the Hybrid Single-Particle Lagrangian Integrated Trajectory Model (HYSPLIT)

(<http://www.arl.noaa.gov/ready/hysplit4.html>)⁸, the air masses...”.

Reference:

8 Stein, A. F. et al. NOAA’s HYSPLIT Atmospheric Transport and Dispersion Modeling System. *Bulletin of the American Meteorological Society* 96, 2059-2077 (2015).

n) Lines 164 to 165. A citation for the CWT method should be given

Response:

Thank you for this comment. In our original manuscript, the CWT method was adapted from “Ying-Kuang Hsua, T. M. H. & Hopke, P. K. Comparison of hybrid receptor models to locate PCB sources in Chicago. *Atmos. Environ.* 37, 545–562 (2003)”. However, in our revised manuscript, CWT results have been removed

because the spatial distributions of CCl₄ emissions have been derived from inverse modeling.

o) Line 235. Suggest “This comparison indicates ... did not fall over this period of time” although, given the inherent problems I have identified with the estimate for the computation of the emission of CCl₄ in Eastern China, I have to also state this sentence may need to be altered in a more significant manner, or perhaps removed.

Response:

Thank you for this comment. Based on your suggestion, we have altered the emission estimation methods to inverse modeling. The inversion results show no decreasing trend of CCl₄ emissions after 2019. Thus, this sentence was revised as follows:

Revised text:

Lines 195-196 in the revised manuscript: *“This comparison indicates that the CCl₄ emissions in eastern China did not fall over this period.”*

p) Line 268. Suggest “Our observations reveal ...”

Response:

Thank you for this comment. We have revised following the suggestion.

Revised text:

Lines 223-224 in the revised manuscript: *“Our observations reveal that MGPM, MRCM, MEMA, and MA sectors are potential sources of CCl₄ emissions...”*.

q) Line 307. In addition to the website, should also provide a reference here, for HYSPLIT.

Response:

Thank you for the comment. We have added the reference of “Stein, A. F. et al. NOAA’s HYSPLIT Atmospheric Transport and Dispersion Modeling System. Bulletin of the American Meteorological Society 96, 2059-2077 (2015)” in the revised manuscript.

r) Line 506. Figure 1 is very important: this figure is the first graphical component of the paper. The figure should be improved, either by zooming in to make the geographic region of China stand out more clearly, of perhaps some other manner.

The three words on panel c are hard to read. Finally, I suggest also pointing out the location of Beijing, so that readers can better orient the locations of ZJU, SHH, and Gossan.

Response:

Thank you for this comment. To clearly show our inversion region, Figure 1 is zoomed in to eastern China, and the boundary of the inversion area is highlighted in bold black. In addition, we show the footprint covering the entire China in Supplementary Fig. 2 and use bold border lines to highlight the Chinese regions. We agree that showing the location of Beijing will help readers understand the location of the ZJU, SHH, and Gosan stations. But instead of showing the location of Beijing to help readers determine where ZJU, SHH, and Gosan are, we provide a map with the name of each province (shown in Supplementary Fig. 1). The revised figures are shown below:

Revised figure:

Fig. 3 Average emission sensitivity derived from FLEXPART simulations and the average emission sensitivity of ZJU and SHH stations to that of the GSN station in 2021 and 2022. a, b Average emission sensitivities from ZJU and SHH stations. c, d Average emission sensitivities from GSN station. e, f Ratio of the average emission sensitivity of ZJU and SHH stations to that of the GSN station. ZJU, SHH, and GSN sites are represented by the blue triangle, dot, and cross, respectively. The area framed by the bold black line is the target area for the inversion. The name of each Chinese province is shown in Supplementary Fig. 1. Distribution of the emission sensitivities for a larger domain than Fig. 1 is shown in Supplementary Fig. 2.

Supplementary Fig. 4 Provinces in China.

Supplementary Fig. 5 Average emission sensitivity derived from FLEXPART simulations and the ratio of the average emission sensitivity between different stations in 2021 and 2022. a Average emission sensitivities from ZJU and SHH stations. **b** Average emission sensitivities from GSN station. **c** Ratio of the average emission sensitivity at ZJU and SHH stations to that at GSN station. Sites at ZJU, SHH, and GSN are represented by the blue triangle, dot, and cross, respectively. The area outlined by the bold black line is China. The name of each Chinese province is shown in Supplementary Fig. 1.

s) Line 518. I have stared at this figure, and can not find either the black or pink dots. If these dots are truly present, they should be made more visible.

Response:

Sorry for the redundant information we provided before. Since spatial distribution of emission hotspots are gained from inverse modeling in our revised manuscript, the figure of CWT was replaced with following plots:

Revised figure:

Fig. 6 Mean Spatial distribution of CCl_4 emissions in eastern China derived from uniformly distributed prior emissions. a For 2021. b For 2022. Spatial distributions of CCl_4 emissions from each inversion using REBS or AGAGE baseline filtering method are shown in Supplementary Fig. 4. Supplementary Fig. 5 is the same as Supplementary Fig. 4 using population-proxy prior emissions.

t) Line 527. Figure 5c is very very important, in my opinion. Figure 5b is easy to “understand”, but hard to decipher, as many of the colors are similar. Perhaps various symbols with a smaller number of colors should be used. Finally, Figure 5a is hard to understand: what is the inset?

Response:

Thank you for this comment. According to the suggestion, different symbols have been used to represent different industry sectors in Figure 5b (Figure 4a in our revised manuscript). The original Figure 5a is unnecessary and has been removed accordingly.

Revised figure:

Fig. 7 Measured CCl₄ concentrations from each industry sector. a Spatial distribution of sampling industry sectors with different symbols. **b** Measured CCl₄ concentrations from each industry sector. Box plots indicate median (middle line), 25th,75th percentile (box) and 5th and 95th percentile (whiskers) as well as the mean level (asterisks). Solid dots represent the concentration of each sample.

END OF REVIEW

Responses to Reviewer #2:

Reviewer #2 (Remarks to the Author):

This manuscript reports on unique measurements of the ozone-depleting CCl₄ from two new sites in China during 2021 and 2022. Quantifying CCl₄ emissions from atmospheric measurements is an important issue worth of consideration by this journal because substantial CCl₄ emissions persist globally and are delaying recovery of the ozone layer, but the sources responsible for these emissions have not been adequately identified and characterized. Ongoing measurements from these new sites have the potential to inform us more accurately about emissions from unique regions of China than has been possible previously. The authors find that emissions from eastern China are comparable in magnitude to those published elsewhere for this same region through 2019 using measurement data from a different site (Gosan, S. Korea). The authors argue that the emissions they have derived for CCl₄ during 2021-2022 are unusually high owing to the “implementation of strict government supervision for CCl₄” after 2018. In an intriguing second part of the paper, the authors report on measurements of CCl₄ from air collected from the chimney vents of 379 businesses that represented 17 different industry sectors. Unusually high CCl₄ concentrations were observed from some sectors that haven’t been associated with CCl₄ emissions in the past. While the data presented are unique and relevant to the issue of ozone depletion and ozone layer recovery, I cannot recommend publication at this time for a number of reasons that I delineate below.

Response:

Thank you for reviewing our work. Below we have responded to and addressed each comment.

First, the importance and uniqueness of this manuscript primarily relates to the availability of new measurement data from new sites AND the assertion that the derived emissions from eastern China are higher than they should be owing to the “implementation of strict government supervision for CCl₄” during this period. However, there is no indication that the purpose of the “supervision program” was to

limit CCl₄ emissions. Certainly, it was instituted to improve tracking of produced CCl₄ so that large amounts couldn't be diverted to produce CFCs, but no indication is provided by the authors to suggest that this program might have had as its goal substantial reductions in CCl₄ emission (and I don't know this independently).

Response:

Thank you for the question. In 2018, an unexpected rise in global trichlorofluoromethane (CFC-11) emission was reported¹. At least 40-60% of this unexpected rise in emissions was attributed to eastern China, with the likely production of CFC-11 and related CCl₄². The Chinese government has launched special inspection law enforcement actions on ODSs (including CFCs, CCl₄, and HCFCs etc.) since 2018.³⁻⁷ The goal of this action is not only to crack down the production and consumption of CFC-11 but also the production and consumption of other regulated ODSs.

To strengthen the regulation of CFC-11 and CCl₄ production, in 2019, the Ministry of Ecology and Environment of China dispatched 228 personnel in 16 working groups to perform on-site supervision for 16 enterprises producing CCl₄ as a by-product in the country⁷. In 2019, CCl₄ online monitoring facilities were installed within these 16 enterprises, and 36 rapid instruments for detecting ODSs in products were dispatched to 22 provinces, strengthening law enforcement and monitoring capabilities across China⁷. By 2021, the number of CCl₄ by-product enterprises under on-site supervision had increased to 19⁸, comprising verifiable and quantifiable CCl₄ online production monitoring systems connected to the central government monitoring platform⁹. The "Supervision program" was implemented to improve tracking of produced CCl₄ to prevent its conversion to CFCs (e.g., CFC-11) or its sale for controlled or emissive applications. Therefore, as the "Supervision program" proceeds, the CCl₄ emissions might decrease.

Revised text:

Lines 91-93 in the revised manuscript: "Additionally, the "Supervision program" was implemented to improve tracking of produced CCl₄ to prevent its conversion to CFCs (e.g., CFC-11) or its sale for controlled or emissive applications."

Reference:

- 1 Montzka, S. A. et al. An unexpected and persistent increase in global emissions of ozone-depleting CFC-11. *Nature* 557, 413-417 (2018).
- 2 Rigby, M. et al. Increase in CFC-11 emissions from eastern China based on atmospheric observations. *Nature* 569, 546-550 (2019).
- 3 People's Government of Jiangxi Province. Jiangxi comprehensively launches ODS special law enforcement inspection (in Chinese), http://www.jiangxi.gov.cn/art/2020/9/14/art_6392_2860755.html (2020).
- 4 Shandong Provincial Department of Ecology and Environment. Announcement from the Shandong Provincial Department of Ecology and Environment on the joint "double random, one public" inspection involving ozone depleting substances (ODS) in 2022 (in Chinese), http://www.sdein.gov.cn/hjjc/zhyw/202211/t20221109_4125354.html (2022).
- 5 Ministry of Ecology and Environment of China. Ecological Environment Law Enforcement Work Plan in Jiading District in 2022 (in Chinese), http://www.jiading.gov.cn/publicity/jcgk/zdgtkwj/gbmwj__publicity/fdzdgnr/zfwj/gbmwj/151518 (2022).
- 6 Ministry of Ecology and Environment of China. Press Conference of the Ministry of Ecology and Environment: Focus on the Prevention and Control of Air Pollution in Autumn and Winter in October (in Chinese), <https://www.solidwaste.com.cn/news/329215.html> (2021).
- 7 Ministry of Ecology and Environment of China. Record of the regular press conference of the Ministry of Ecology and Environment in August 2019 (in Chinese), https://www.mee.gov.cn/xxgk2018/xxgk/xxgk15/201908/t20190830_730891.html (2019).
- 8 Ministry of Ecology and Environment of China. Progress of ecological and environmental law enforcement work and next work arrangements (in Chinese), <https://baijiahao.baidu.com/s?id=1698552840313072252&wfr=spider&for=pc> (2021).
- 9 Ministry of Ecology and Environment of China. Record of the regular press conference of the Ministry of Ecology and Environment in April (in Chinese),

https://www.mee.gov.cn/xxgk/2018/xxgk/xxgk15/202104/t20210428_831177.html

(2021).

Second, the emission derivation is performed here in a manner that substantially underestimates the true uncertainties, making me question the reliability of the CCl₄ emission estimates provided. Emissions of CCl₄ in this work are derived by scaling the concentration enhancements for CCl₄ to those measured concurrently for CHCl₃ and multiplying that ratio by the “known” CHCl₃ emission. This method works best if the enhancements for the two gases are highly correlated. Figure 2 of the Supplementary material shows the correlation, but its significance isn't quantified or mentioned (and visually does not look very high). Most worrisome, however, is that emissions of CHCl₃ from eastern China from 2013 (48.3 Gg) and 2015 (80-95 Gg) (not 2021 or 2022) are used to derive emissions for CCl₄ in 2021 and 2022 based on measurements in this latter years. No consideration of what CHCl₃ emissions might have been in 2021 or 2022 is mentioned, making the estimate of CCl₄ emissions in the key years highly suspect, particularly since the large difference (change?) in CHCl₃ emissions during 2013 and 2015 suggests that CHCl₃ emissions from eastern China can change substantially from year-to-year.

Response:

Thank you for this comment. We highly agree with this comment, which is also pointed out by reviewer #1. There is indeed a large uncertainty (or poor reliability) in estimating emissions using the interspecies correlation method regarding the poor correlation (Pearson's $r=0.29$) between CCl₄ and CHCl₃ and other issues pointed out in the comment. In our revised manuscript, we have switched to the inverse modeling method instead of the interspecies correlation method, avoiding the flaws in the interspecies correlation method, which is also suggested by reviewer #1. The details of the inverse modeling method description and figures are below.

Revised text:

Lines 269-338 in the revised manuscript: “*This study used an inverse modeling technique based on a FLEXPART atmospheric transport simulation model and the Bayesian inversion algorithm to quantify CCl₄ emissions. The FLEXPART-based inversion method has been applied in many previous studies¹⁰⁻¹². In brief, driven by the meteorological data (European Centre for Medium-Range Weather Forecasts) with a spatial resolution of $1^\circ \times 1^\circ$ and a temporal resolution of 3 h, the FLEXPART*

model was ran in backward mode for 20 days. The source–receptor relationship (termed as “emission sensitivity”) matrix was established based on the backward simulation. Combining the derived emission sensitivity matrix, Bayesian inversion technique, and the CCl₄ data from ZJU and SHH, yielded the CCl₄ emission strength (1° × 1°) in grid cells over eastern China. The associated equations are as follows:

$$J(x) = \frac{1}{2} (x - x_a)^T S_a^{-1} (x - x_a) + \frac{1}{2} (y^{obs} - Hx)^T S_o^{-1} (y^{obs} - Hx)$$

By solving $\nabla_x J(x) = 0$ yields:

$$x = x_a + S_a H^T (H S_a H^T + S_o)^{-1} (y^{obs} - Hx_a)$$

$$S_b = (H^T S_o^{-1} H + S_a^{-1})^{-1}$$

where x represents the state vector of the emission strength, y^{obs} represents the CCl₄ measurement vector, x_a represents the prior emission vector, H is the emission sensitivity, S_a and S_b are the error covariance matrix of prior and posterior emissions, respectively, and S_o represents the error covariance matrix of measurement data. To obtain the prior emission vector (x_a), the national total CCl₄ emissions in China during 2011–2015 (23.6 ± 7.1 Gg yr⁻¹)¹³ and emissions in other countries (derived from global emissions [44 ± 15 Gg yr⁻¹ in 2020]¹⁴ and emission estimates of the United States [4.0 (2.0–6.5) Gg yr⁻¹ during 2008–2012]¹⁵, Japan [0.6 Gg yr⁻¹ in 2014]¹⁶, South Korea [0.2 Gg yr⁻¹ in 2014]¹⁶, and India [2.8 Gg yr⁻¹ in 2014]¹⁶) were assigned to grid cells with a uniform spatial distribution in eastern China, and a population-proxy distribution in other regions (Supplementary Fig. 9a). The inversion results were also evaluated using prior emissions following population distribution in 2020¹⁷ (Supplementary Fig. 9b).

The robust extraction of baseline signal (REBS) method¹⁸ was applied to distinguish background and non-background concentrations of the in-situ CCl₄ concentrations at ZJU. The REBS is a statistical method developed by Ruckstuhl et al.¹⁸ to extract background signals using a robust local regression model, and has been widely applied to determine baselines of trace gases in inversion studies^{19,20}. The observed concentration at a certain time ($y(t_i)$) was divided into three parts as shown

depicted in the following equation:

$$y(t_i) = g(t_i) + m(t_i) + e_i$$

where $g(t_i)$ represents the background concentration at time t_i , $m(t_i)$ is the enhanced concentration caused by polluted plum during t_i , and e_i represents the observational error.

The baseline curve g was obtained using the REBS technique over a sufficiently long bandwidth (90 days) by assuming that most observations are at background levels and that the baseline signal changes slowly relative to the regional signal. In this method, data points closer to the time of consideration were given more weight, and data points outside a specific range (1.5σ in this study) were iteratively excluded.

For flask samples at SHH, the background concentration of CCL_4 was determined as the lowest concentration measured in a two-month moving window. The observational error (σ_{obs}) was calculated as follows:

$$\sigma_{obs} = \sqrt{\sigma_{obs_precision}^2 + \sigma_{obs_representation}^2 + \sigma_{background}^2}$$

where $\sigma_{obs_precision}$ is the measurement precision of CCL_4 , $\sigma_{obs_representation}$ stands for representation of the observation, $\sigma_{background}$ represents the background uncertainty. In this study, $\sigma_{obs_representation}$ was calculated as 1-Sigma standard deviation of the measurements each day for online samples and as 1-Sigma standard deviation of the measurements in a two-month moving window for flask samples. The $\sigma_{background}$ was estimated as 1-Sigma standard deviation of the fitted background concentrations during the sampling period. The diagonal elements of S_o were set to squared σ_{obs} . The diagonal elements of S_a were calculated as squared uncertainty of the prior emission field. The off-diagonal elements of S_a were calculated based on previous studies^{12,21}, with the spatial decorrelation length scale set to 400 Km. In this study, three sets of prior emissions were established (i.e., 150%, 100% and 50% of the reference prior emissions), and three uncertainties (600%, 450%, 300% for uniformly distributed prior emissions) were set under each set of prior emissions. A total of nine

inversions were carried out for each year. The final posterior emissions were the average of the nine inversions (Supplementary Fig. 10). The posterior CCl₄ emissions in eastern China under the nine inversions in 2021 and 2022 were loaded in the range of the three sets of prior emissions (Supplementary Fig. 10). The posterior emissions were insensitive (varied < 9%) to variations in emission uncertainty, suggesting that the prior emissions used in this study were not systematically high or low, and that the observations constrain posterior emissions well. We also tested baselines from the AGAGE baseline filtering method (baseline obtained by fitting the daily minimum with a second-order polynomial)²² to examine the impact of different baselines on posterior emissions. The inversion ensembles (Supplementary Table 3) using two baselines (AGAGE and REBS methods) and two prior emission fields (uniform and population-proxy distributions) show that the posterior emissions varied by < 6% (standard deviation divided by mean) during 2021–2022 (Supplementary Fig. 11).”

Revised figure:

Supplementary Fig. 9 CCl_4 prior emissions used in this study. a Prior emissions in eastern China set to a uniform distribution. **b** Prior emissions in eastern China distributed based on population density. The blue triangle and dot represent the ZJU and SHH sites, respectively.

Supplementary Fig. 10 Posterior CCl_4 emissions in eastern China. *a* Nine inversions (three magnitudes of 150%, 100%, and 50% of reference prior emission multiplied by the three magnitudes of 600%, 450%, and 300% of reference prior emission uncertainty) for CCl_4 emissions in eastern China in 2021 and 2022 based on the baseline extracted with the REBS method and uniformly distributed prior emissions (Inv-REBS-UniformPrior). *b* Same as *a* but based on the baseline extracted with the method suggested by AGAGE (Inv-AGAGE-UniformPrior). *c* Same as *a* but based on the non-uniformly distributed prior emissions (Inv-REBS-PopPrior). *d* Same as *c* but based on the baseline extracted with the method suggested by AGAGE (Inv-AGAGE-PopPrior). The three green horizontal solid lines represent three magnitudes of 150%, 100%, and 50% of reference prior emissions for eastern China.

Supplementary Fig. 8 *CCl₄ emissions from four inversion sets for eastern China.* Detailed information on four inversion sets using two baselines (AGAGE and REBS methods) and two prior emission fields (uniform and population-proxy distributions) is shown in Supplementary Table 3.

Supplementary Table 3 *The set-up for the four inversion frameworks*

Category	Baseline extraction method	Priori emission distribution
Case A	REBS	Uniformly distributed in eastern China, and distributed based on population density in other areas
Case B	AGAGE	Uniformly distributed in eastern China, and distributed based on population density in other areas
Case C	REBS	Distributed based on population density
Case D	AGAGE	Distributed based on population density

Reference:

10 Fang, X. et al. Rapid increase in ozone-depleting chloroform emissions from China. *Nature Geosci.* 12, 89-93 (2019).

11 Fang, X. et al. Changes in HCFC Emissions in China During 2011–2017. *Geophys. Res. Lett.* 46, 10034-10042 (2019).

12 Fang, X. et al. Multiannual top-down estimate of HFC-23 emissions in East Asia. *Environ. Sci. Technol.* 49, 4345-4353 (2015).

13 Park, S. et al. Toward resolving the budget discrepancy of ozone-depleting carbon

tetrachloride (CCl₄): an analysis of top-down emissions from China. *Atmos. Chem. Phys.* 18, 11729-11738 (2018).

14 World Meteorological Organization (WMO). *Scientific Assessment of Ozone Depletion: 2022*, GAW Report No. 278. 509 (World Meteorological Organization (WMO), Geneva, 2022).

15 Hu, L. et al. Continued emissions of carbon tetrachloride from the United States nearly two decades after its phaseout for dispersive uses. *Proc. Natl. Acad. Sci. U. S. A.* 113, 2880-2885 (2016).

16 Sherry, D., McCulloch, A., Liang, Q., Reimann, S. & Newman, P. A. Current sources of carbon tetrachloride (CCl₄) in our atmosphere. *Environ. Res. Lett.* 13, 024004 (2018).

17 Center for International Earth Science Information Network - CIESIN - Columbia University. (NASA Socioeconomic Data and Applications Center (SEDAC), Palisades, New York, 2018).

18 Ruckstuhl, A. F. et al. Robust extraction of baseline signal of atmospheric trace species using local regression. *Atmospheric Measurement Techniques* 5, 2613-2624 (2012).

19 Affolter, S. et al. Assessing local CO₂ contamination revealed by two near-by high altitude records at Jungfrauoch, Switzerland. *Environ. Res. Lett.* 16 (2021).

20 Western, L. M. et al. A renewed rise in global HCFC-141b emissions between 2017–2021. *Atmos. Chem. Phys.* 22, 9601-9616 (2022).

21 Brunner, D. et al. An extended Kalman-filter for regional scale inverse emission estimation. *Atmos. Chem. Phys.* 12, 3455-3478 (2012).

22 S. O'Doherty et al. In situ chloroform measurements at Advanced Global Atmospheric Gases Experiment atmospheric research stations from 1994 to 1998. *J. Geophys. Res.* 106, 20429-20444 (2001).

Third, the authors also assert that they have found CCl₄ emissions in new regions of China (Jiangxi and Fujian provinces). They also acknowledge that the trajectory method can often erroneously place emissions upwind of the actual source region (my

interpretation of the “ghost” sources in the ocean discussed in Figure 3). This seems also a possibility for the proposed CCl₄ emissions these two regions and would need further investigation before one could reliably assert that there are significant emissions coming from these provinces.

Response:

Thank you for this comment. In our revised manuscript, the spatial distribution of CCl₄ emissions was reanalyzed based on the inverse model method, which performed much better than the CWT method in locating emission source regions and could largely avoid the flaw of “ghost” sources.²³ (please see Fig. 3 in our revised manuscript below).

Revised text:

Lines 151-155 in the revised manuscript: *“The northeastern region of Shandong, eastern region of Jiangsu, and southeastern region of Zhejiang were identified in this study as major CCl₄ source regions in eastern China during 2021–2022 (Fig. 3, Supplementary Fig. 4 and Supplementary Fig. 5). Additionally, air mass corresponding to high CCl₄ concentrations typically pass through these regions (Supplementary Fig. 6, see Supplementary Discussion 2 for more details).”*

Revised figure:

Fig. 9 Mean Spatial distribution of CCl_4 emissions in eastern China derived from uniformly distributed prior emissions. a For 2021. b For 2022. Spatial distributions of CCl_4 emissions from each inversion using REBS or AGAGE baseline filtering method are shown in Supplementary Fig. 4. Supplementary Fig. 5 is the same as Supplementary Fig. 4 using population-proxy prior emissions.

Supplementary Fig. 4 Spatial distribution of CCl₄ emissions in eastern China derived from uniformly distributed prior emissions. a, b Based on baseline extracted with REBS method. **c, d** same as a, b, but the baseline is extracted with the method used by AGAGE. The blue triangle and dot represent the ZJU and SHH sites, respectively.

Supplementary Fig. 10 Spatial distribution of CCl₄ emissions in eastern China derived from population-proxy distributed prior emissions. a, b Based on baseline extracted with REBS method. **c, d** same as **a, b**, but the baseline is extracted with the method used by AGAGE. The blue triangle and dot represent the ZJU and SHH sites, respectively.

Supplementary Fig. 6 Clusters of backward trajectories during the sampling period and corresponding CCl₄ concentration. a Cluster analysis of 120 h backward trajectories for ZJU during the sampling period calculated using the HYSPLIT model, with the starting height at 100 m above ground level. Running intervals were set as 1 h for each day; the ratio, moving height, and average concentrations of CCl₄ of each cluster are also presented. **b** Same as (a), excluding the SHH station.

Reference:

23 Fang, X. et al. Performance of Back-Trajectory Statistical Methods and Inverse Modeling Method in Locating Emission Sources. ACS Earth and Space Chemistry 2, 843-851 (2018).

Fourth, in a highly unique section of the paper, concentration measurements were made from the stacks of many industries and potentially suggest the discovery of previously unrecognized CCl₄ sources. However, this result needs some further development before such assertions can be made with any reliability. Questions needing to be explored so that the implications of the new results can be better understood include: how extensive is this industry and how many businesses related to this industry exist in China? What are the potential fluxes of CCl₄ from these stacks? Can you put those two numbers together to estimate potential emissions from this sector? What are the potential underlying variables affecting the range of CCl₄ concentration in stacks associated with the same industry? Particularly MGPM? For this portion of the paper to be useful to a wider audience, it will require some further estimation of these quantities, or reasonable limits applied to them, so that the potential importance of these other sources can be roughly assessed.:

Response:

Thank you for the suggestion.

- (1) As for “how extensive is this industry, and how many businesses related to this industry exist in China?”

The number of enterprises above the designated size (i.e., annual revenue >20 million CNY) for the industry of MGPM and MEMA shows an apparent upward trend in the number of enterprises since 2018. The number of enterprises in most other industries shown in Supplementary Fig. 7 is relatively stable during 2011–2021²⁴, but the overall trend is still rising.

The extensive of each investigated industry was discussed in lines 236-242

in the revised manuscript: “*The number of enterprises with annual revenue > 20 million CNY has exhibited an obvious upward trend for the MGPM and MEMA industries since 2018 (growth rate: 10–16% yr⁻¹ during 2018–2021)²⁵ (Supplementary Fig. 8). Although the number of enterprises in most other industries was relatively stable during 2011–2021²⁴ (Supplementary Fig. 8), the*

overall trend has been on the rise (Supplementary Fig. 8). Considering that they may represent potential sources of CCl₄ emissions, attention should be paid to these industries (particularly MGPM) to evaluate their impact on the ozone layer.”

(2) As for “What are the potential fluxes of CCl₄ from these stacks? Can you combine those two numbers to estimate potential emissions from this sector?” CCl₄ concentrations were only measured in stacks sampled in this study. However, in order to estimate fluxes of CCl₄ from stacks across China, we need to measure CCl₄ concentrations in all stacks across China (or major stacks across China) and the exhaust flow in the corresponding stack while these data are unavailable. Future studies are encouraged to measure the concentrations and flux in more enterprises.

(3) As for “What are the potential underlying variables affecting the range of CCl₄ concentration in stacks associated with the same industry?”

We explore the relationship between revenue and measured concentration of CCl₄ to discuss the potential underlying variables affecting the range of CCl₄ concentration in stacks.

The relationship between revenue and CCl₄ concentrations in stacks was discussed in lines 230-235 in the revised manuscript: *“Based on the results of this study, high concentrations of CCl₄ in 17 industries were observed in enterprises with low annual revenue (Supplementary Fig. 7). Specifically, the CCl₄ concentration was negatively correlated with the logarithm of the enterprise revenue ($R^2 = 0.49$, $p < 0.01$). Similar relationships between CCl₄ concentration and enterprise revenue were also observed in the MGPM ($R^2 = 0.75$, $p < 0.01$) and MRCM ($R^2 = 0.49$, $p < 0.01$) industries.”*

Revised figure:

Supplementary Fig. 7 Relationship between CCl_4 concentrations and revenue. a CCl_4 concentrations across revenue ranges for all sampled enterprises. **b.** Same as **a**, excluding MGPM. **c.** Same as **a**, excluding MRCM. The size of the dots represents the number of enterprises.

Supplementary Fig. 8 Number of enterprises above designated size (annual revenue > 20 million CNY) for each industry²⁴.

Reference:

24 National Bureau of Statistics (NBS). China Statistical Yearbook. (China Statistics Press, 2011-2021).

25 National Bureau of Statistics (NBS). China Statistical Yearbook. (China Statistics Press, 2021).

Responses to Reviewer #3:

Reviewer #3 (Remarks to the Author):

Review of the manuscript: Substantial carbon tetrachloride emissions in eastern China, after 2010 dispersive-use phase out and 2019–2022 strict government supervision: A concern for the ozone layer by Li et al., submitted to Nature Comm. The manuscript is related to on-going emissions of CCl₄, for which a consistent gap between bottom-up and top-down emissions exists globally, with higher emissions estimated by measurement-based top-down methods. The manuscript discusses enhanced CCl₄ concentrations in eastern China and estimates emissions, which are higher than expected. Authors also make use of industrial samples, which show massively high concentrations of CCl₄ and related them to certain types of industries. I am in favour of publishing this manuscript in Nature Comm. because of its global implications. However, some of the comments should be covered carefully.

Response:

Thank you for your support on this work. Below, we have responded to each of the comments.

General comments

1. The most important finding is the massive concentrations close to industrial sites. Here an explanation with a high grade of probability is forbidden emissive uses of CCl₄. Authors should clearly state this in the manuscript. This would be a very good point to make and this would also close the gap between the bottom-up and the top-down estimate, as in the bottom-up numbers such illegal practices are simply not taken into account as it was deemed highly unlikely, but obviously not.

Response:

Thank you for this comment.

We acknowledge that (1) high concentrations of CCl₄ may indicate forbidden dispersive uses of CCl₄, (2) high concentrations of CCl₄ may be formed as a byproduct during the use or production of chemicals/products, (3) high concentrations of CCl₄ come from using CCl₄ as feedstock to produce other chemicals/products. Due

to the lack of definite evidence, we cannot judge which one or two or three of the above three causes of high concentrations of CCl₄. Therefore, we cannot clearly state that there is illegal use of CCl₄. Based on your suggestion, we have added the possible explanations for the high concentration from industrial sectors in our revised manuscript.

Revised text:

Possible explanation for the high CCl₄ concentrations was stated in lines 220-223

in the revised manuscript: *“The high CCl₄ concentrations in these sectors may arise from 1) byproducts of CCl₄ during the use or production of chemicals/products, 2) using CCl₄ as feedstock to produce other chemicals/products, or 3) other unidentified uses of CCl₄.”*

2. Accordingly, point number 1 should also be mentioned in the abstract. At the moment the paper stays very vague on this. The readers want to know the reason for these emissions and it does not make it better to hide it away.

Response:

We thank the reviewer for the comments. We agree that revealing the massive concentrations measured at industrial sites is important. Our revised abstract clearly presents the MGPM and MRCM sectors of a potential industrial source of CCl₄. As for the causes of CCl₄ emissions. However, since there is still no conclusive evidence to prove whether there are prohibited use of CCl₄ (please see our responses in point number 1), we only objectively reveal the high concentrations and sectors.

Revised text:

Lines 28-31 in the revised manuscript: *“Subsequently, our study identified potential industrial sources (manufacture of general purpose machinery and manufacture of raw chemical materials, and chemical products) of CCl₄ emissions that were not revealed in previous studies and were not under governmental supervision.”*

3. Several citations are at the edge of misquotation. These will be detailed in the specific comments below.

Response:

Thank you for this comment. In our revised manuscript, the misquotation has been corrected, and we have responded to the specific comments below.

4. Data must be openly accessible.

Response:

The emission sensitivity data sets from backward simulations for each station in this study are available through XXXXXX URL. Measurement data are available from the corresponding author upon reasonable request.

Specific comments:

Line 21

...was globally phased out by 2010, including China.

Response:

We thank the reviewer for this comment. The sentence has been revised as follows:

Lines 20-22 in the revised manuscript: *“According to the Montreal Protocol, production and consumption of ozone-layer-depleting CCl₄ for dispersive applications, was globally phased out by 2010, including China.”*

Line 23

...were disclosed, and the latest...

Response:

We thank the reviewer for this comment. The sentence has been revised as follows:

Lines 22-23 in the revised manuscript: *“However, continued CCl₄ emissions were disclosed, with the latest CCl₄ emissions unknown in eastern China.”*

Line 27

...respectively, well above the global background.

Response:

We thank the reviewer for this comment. This sentence has been removed as the concentration discussion has been shortened as suggested by the reviewer.

Line 29

In 2010 and after the implementation...

Response:

Thank you for the suggestion; the sentence has been revised accordingly:

Lines 25-27 in the revised manuscript: *"This indicates that CCl₄ emissions*

continued after being phased out for dispersive uses in 2010 and after the implementation of strict government supervision...."

Line 42

Feedstock instead of "raw material"

Response:

Thank you for flagging this; the sentence has been revised accordingly:

Line 42 in the revised manuscript: *"An exemption was in its use as a chemical feedstock or processing agent."*

Line 42

Ref 6 is not needed for this argument and does not really cover this

Response:

Thank you for suggestion. Ref 6 has been deleted in our revised manuscript.

Line 45

[8] is wrongly cited. In the report they say:

Global CCl₄ emissions did not significantly decline during the 2010–2019 period (0.1 ± 0.2 Gg yr⁻¹), 41 ± 3 and 38 ± 3 are not mentioned in the report (or at least this reviewer has not seen it)

Response:

Thanks, 41 ± 3 and 38 ± 3 were extracted from estimate using AGAGE in Figure 3.4. in the *"Report on unexpected emissions of CFC-II"* (shown below). Following your suggestion, the original sentence has been revised to *"Top-down estimates of global emissions, inferred from atmospheric observations, have shown no significant decrease in CCl₄ emissions between 2010 and 2019 (decline rate of 0.1 ± 0.2 Gg*

yr⁻¹)”.

Lines 42-45 in the revised manuscript “*Top-down estimates of global emissions, inferred from atmospheric observations, have shown that there was no significant decrease in CCl₄ emissions between 2010 and 2019 (decline rate of 0.1 ± 0.2 Gg yr⁻¹)”.*

Line 47

Take the newest numbers from the most recent ozone assessment (2022)

Response:

Thank you for the comment. The original numbers have been replaced with the newest numbers from the most recent ozone assessment (2022).

Lines 46-47 in the revised manuscript: “*...approximately 26.1 Gg (12.6–40.0 Gg) emissions in 2019, 17 Gg yr⁻¹ lower than those of the top-down emission estimates (43 Gg yr⁻¹ in 2019)¹.*”

Reference:

1 World Meteorological Organization (WMO). Scientific Assessment of Ozone Depletion: 2022, GAW Report No. 278. 509 (World Meteorological Organization (WMO), Geneva, 2022).

Line 54

...CCl₄ emissions from dispersive uses were reduced...

Response:

Thank you for the suggestion; the sentence has been revised accordingly:

Lines 53-54 in the revised manuscript: “... *CCL₄ emissions from dispersive uses were reduced from 9.3 Gg yr⁻¹ in 2005 to 1.1 Gg yr⁻¹ in 2010 and to zero afterwards.*”

Line 55

A bottom-up inventory for non-emissive uses of CCl₄, including...

Response:

Thank you for the comment; the sentence has been revised accordingly:

Lines 54-55 in the revised manuscript: “*a bottom-up inventory for non-emissive CCl₄ uses, including coal combustion, tetrachloroethylene conversion,...*”

Line 59

Moreover, another independent study showed bottom-up emission estimates...

Response:

Thank you for the suggestion; the sentence has been revised as follows:

Lines 58-59 in the revised manuscript: “*Moreover, another independent study showed bottom-up emission estimates of CCl₄ in 2014 of 7.3 Gg yr⁻¹...*”

Line 81

Rigby et al say at least 40-60%

Response:

Thank you for the suggestion. The sentence has been revised as follows:

Lines 80-81 in the revised manuscript: “*At least 40–60% of this unexpected increase was attributed to eastern China with the likely production of CFC-11 and related CCl₄.*”

Line 178

Equation 4 (see methods)

Response:

Thank you for pointing this out. Since we switched to inverse modeling for estimating CCl₄ emissions, the description of the interspecies correlation method has been removed.

Line 187

Reference missing for REBS

Response:

Thank you for pointing this out; we have now added the reference “*Ruckstuhl, A. F. et al. Robust extraction of baseline signal of atmospheric trace species using local regression. Atmospheric Measurement Techniques 5, 2613-2624 (2012).*”

Lines 20-22 in the revised Supplementary information: “*Using the robust extraction of baseline signals (REBS) method², approximately 30% of the CCl₄ concentrations at ZJU (averaged as 93.9 ± 14.4 ppt) were identified as non-background concentrations, respectively.*”

Reference:

2 Ruckstuhl, A. F. et al. Robust extraction of baseline signal of atmospheric trace species using local regression. *Atmospheric Measurement Techniques* 5, 2613-2624 (2012).

Line 188

...for total China reported...

Response:

Thank you for the suggestion; the description has been revised as follows:

Lines 136-138 in the revised manuscript: “*the CCl₄ emissions in eastern China reported in this study... are lower than those for total China reported by Park et al.³ (23.6 ± 7.1 Gg yr⁻¹ during 2011–2015) ...*”

Reference:

3 Park, S. et al. Toward resolving the budget discrepancy of ozone-depleting carbon tetrachloride (CCl₄): an analysis of top-down emissions from China. *Atmos. Chem. Phys.* 18, 11729-11738 (2018).

Line 203

...from emissive uses...

Response:

Thank you for pointing this out; the description has been revised accordingly:

Line 165 in the revised manuscript: *“The CCl₄ emissions from emissive uses in China were estimated to be zero....”*

Line 214

The relative importance...

Response:

Thank you for the suggestion; the description has been revised accordingly:

Line 175 in the revised manuscript: *“The relative importance of CCl₄ emissions is increasing in China.”*

Line 375

Data have to be open access

Response:

Thank you for the suggestion; the description has been revised as follows:

“The emission sensitivity data sets from backward simulations for each station are available through XXXXXX url. Measurement data are available from the corresponding author upon reasonable request.”

Reviewer #1 (Remarks to the Author):

Please see attached PDF.

Reviewer #1 Attachment on the following page

Second review of NCOMMS-23-29246A, Substantial carbon tetrachloride emissions in eastern China, after 2010 dispersive-use phase-out and 2019–2022 strict government supervision: A concern for the ozone layer, by Li et al.

The paper is very much improved. The authors have addressed all of my major concerns regarding methodology and the citation of appropriate, prior studies.

I still question whether the primary result of continued, high emissions of CCl₄ from China (that is, the two red diamonds in Figure 2) are worthy of being so much of the focus of a paper in this journal, given the prior literature on this topic. However, the analysis is sound, the dataset is new and unique, and the continued emission of CCl₄ from China does indeed pose a threat to the ozone layer.

I have a number of minor comments:

- a) assuming the paper is accepted, please try very hard, between now and submission of the final manuscript for production of galleys, to remove each instance of the word “This indicates” being used as a noun. For example, the sentence “This indicates ...” in line 26 reads so much better as “This finding indicates ...”
- b) the sentence on lines 117 to 121 is longer than needed; suggest ending at after “Fig. 2)” on line 120.
- c) Line 140: text would read better as “are lower than the total emission for all of China”. On line 139, perhaps “emissions” would be “emission”.
- d) Lines 145 & 147: in my opinion, the paper is easier to digest if “CCl₄” were not used as an adjective, so often. Here, I suggest “consumption of CCl₄”
- e) Line 157: perhaps “air masses”
- f) Line 159: here, “high emissions of CCl₄” reads much better in my opinion
- g) Line 161: how about “indisputable emissions of CCl₄” (see <https://forum.wordreference.com/threads/indisputable-and-undisputed.1474726> for some thoughts in undisputable versus indisputable
- h) Line 169: perhaps a comma after “Wan et al.”
- i) Line 177: suggest “associated emission gaps for CCl₄”.
- j) Line 185: no need in my opinion to repeat the value of 6.6 +/- 2.7 given just above. This sentence is still long and a bit awkward even this is deletion. Perhaps the term CFC-11-eq Gg yr⁻¹ can be given once, early in the sentence, so that the reader could more easily focus on the numbers and various species. Finally, I suppose “(but for all of China)” applies only to HCFC-22; the sentence could be modified to that there is clarity about how this phrase should be applied.
- k) Line 206: suggest “potential sources of CCl₄”
- l) Line 207: suggest “This campaign” rather than “This study” to connect better to prior sentence
- m) line 225: suggest “high concentrations of CCl₄”.

n) lines 235 to 240: not sure what the connection of high concentrations of CCL4 with low annual ravenous enterprises means. I wonder if either another sentence could be added explaining why the authors think this could be important, or else perhaps dropping this paragraph and the associated figures from Supplement. If the editor is reading this deep into the minor comments, and wants to shorten the paper, I think not only this paragraph but also lines 241 to 248 could be removed.

o) Lines 249 to 252: Sorry, I think this is a rather weak concluding paragraph. Here, I would add in the years of the observations, as well as words in plain language (not abbreviations) of the most egregious industrial sources of CCL4.

q) The station location on Figure 1 was a bit hard to see based on a printed copy of the paper. I could only understand by enlarging the PDF. Also, the caption uses "cross" that I interpret to be "+", rather than "X".

In the interest of getting this back to the journal by the deadline, coupled with the holiday in my home country, I shall admit I did not review the Methods section in the revised paper. I did however read the responses to my prior comments carefully (I was reviewer 1), and I think the use of FLEXPART to compute the emission of CCL4 rather than the correlation method is a major step forward. I have no concerns about the methodology of the revised paper.

END OF REVIEW

Reviewer #2 (Remarks to the Author):

It remains true that on a global scale substantial gaps still exist in our understanding of the measured global atmospheric decline of CCl₄ concentrations. Emissions derived from global atmospheric concentration changes (the top-down approach) are larger than can be explained by activity-based estimates of sources (bottom-up approach). Measurements of CCl₄ in regions dense with chemical production plants, especially in China, are highly valuable and have the potential to improve our understanding of these important questions regarding CCl₄ emissions. As a result, this work reporting high quality and high-frequency measurements at two new sites in China has the potential to add significantly to our understanding of these gaps and of the magnitude and nature of CCl₄ emissions coming from this important region of the world.

The authors have made substantial improvements to their manuscript in response to reviewers' comments, especially in the methods used to derive emissions. In a few instances, however, they failed to address fundamental concerns expressed in the initial review (e.g., accurately conveying the goal of the "MEE supervision program", was it for better tracking of production and not, as the authors suggest, related to minimizing residual CCl₄ emissions). This point is central to how one might expect emissions from known processes to have evolved over time. As a result, I disagree with their central conclusion about the presence of a substantial gap in our understanding of CCl₄ emissions from China in the most recent years (2021-2022). In previous years, there clearly were substantial unexpected and unexplained CCl₄ emissions, and this point has been made in many previous studies. But the results presented here for 2021-2022, which give a better picture of total Chinese emissions than previous studies owing to the location of the new sites, suggest that there is no longer a gap in our understanding of Chinese emissions in this recent period. This is essentially the opposite message being conveyed in the current manuscript. Without substantial revision of the main points of this paper I cannot condone publication in any Journal.

Bottom-up analyses encompassing all known sources not controlled by the Montreal Protocol and, in my estimation, not likely to decrease over time are indicated as contributing 7.3 Gg/yr of emission in 2014 (lines 56-61). Because the MEE supervision of CCl₄ production that was adopted after 2018 was focused primarily on tracking CCl₄ production in order to avoid any siphoning of this production to create CFCs, it seems unlikely to expect that it had any effect on residual, leakage-related emissions from non-controlled sources. The authors have implied that the MEE program should have caused emission reductions, but they failed to provide any evidence to back up this suggestion in their response. Hence, I have no reason to expect substantial declines in CCl₄ emissions in the years following 2014 from uncontrolled processes; in fact, I expect that they could have increased in recent years, given the use of CCl₄ as feedstock to produce HFCs in larger amounts in recent years.

The new results presented in this manuscript, which the authors rightly suggest provide a more complete picture of total emissions from China than studies relying on data only from Gosan (see Figure 1), yield a CCl₄ emission of 7.0-8.2 Gg/yr in 2021 and 2022, with uncertainties of about 3 Gg on each value. Given the consistency of these emissions magnitudes with expectations from uncontrolled sources (7.3 in 2014, and perhaps increasing after), it seems reasonable to conclude that there is no longer a gap in our understanding of CCl₄ emissions from China.

The additional findings discussed in this manuscript related to previously unidentified sources of CCl₄ is interesting and needs further study, but the evidence presented here suggests that those unaccounted sources are likely small relative to Chinese emission totals.

In some sections of the manuscript, the authors suggest that the new measurements provide only estimates for eastern China, and that some earlier work (Park et al., 2018 and Lunt et al., 2018) argues for total Chinese emissions being substantially larger. But I do not think this is an accurate interpretation. The earlier studies cited by the authors are based on very uncertain extrapolations of measurements from the Gosan station, which the authors have clearly demonstrated to be (Figure 1) much less likely to represent emission magnitudes from China more broadly than the results from their new stations.

In summary, this paper provides important new measurements data and better estimates of total Chinese CCl₄ emissions, and the new results are likely more accurately interpreted given the available information to suggest that our understanding of CCl₄ emissions from China have improved over time, not the opposite. What would strengthen the manuscript notably is a very close look at the bottom-up emission estimates from non-controlled processes to understand how they have evolved over time instead of presuming without careful consideration what influence the "supervision program" might have had on emission expectations for CCl₄ in years since 2018.

Responses to Reviewer #1:

Second review of NCOMMS-23-29246A, Substantial carbon tetrachloride emissions in eastern China, after 2010 dispersive-use phase-out and 2019–2022 strict government supervision: A concern for the ozone layer, by Li et al.

The paper is very much improved. The authors have addressed all of my major concerns regarding methodology and the citation of appropriate, prior studies.

Response:

Thank you for your suggestion to improve the quality of our manuscript. Below, we have responded to each of the comments.

I still question whether the primary result of continued, high emissions of CCl₄ from China (that is, the two red diamonds in Figure 2) are worthy of being so much of the focus of a paper in this journal, given the prior literature on this topic. However, the analysis is sound, the dataset is new and unique, and the continued emission of CCl₄ from China does indeed pose a threat to the ozone layer.

Response:

Thank you for this comment.

We agree with the reviewer that previous studies have reported high CCl₄ emissions in China. However, this study has significantly advanced this research field because: (1) we carried out CCl₄ concentration observations that could better constrain emissions in eastern China than at the site of GSN used in previous studies; (2) we focused on the latest period of 2021–2022 in which CCl₄ emissions may have changed; (3) we provide the latest top-down estimates of CCl₄ emissions in eastern China which are lacking in all previous studies; and (4) most importantly, we identified potential new sources of CCl₄ emissions to better minimize the gaps between the top-down and bottom-up emissions.

We thank the reviewer for the comment that “the analysis is sound, the dataset is new and unique, and the continued emission of CCl₄ from China does indeed pose a threat to the ozone layer”.

I have a number of minor comments:

Response:

Thank you. We have responded to each comment below.

a) assuming the paper is accepted, please try very hard, between now and submission of the final manuscript for production of galleys, to remove each instance of the word “This indicates” being used as a noun. For example, the sentence “This indicates ...” in line 26 reads so much better as “This finding indicates ...”

Response:

We thank the reviewer for the suggestion. The “This indicates” in the manuscript was revised as “This finding indicates ...”.

b) the sentence on lines 117 to 121 is longer than needed; suggest ending at after “Fig. 2)” on line 120.

Response:

We thank you for the suggestion. The sentence after “Fig. 2)” has been removed, and the original sentence was revised as below:

Lines 111–114 in the revised manuscript: “*Based on the emission sensitivity derived from the backward simulation using the FLEXible PARTicle dispersion (FLEXPART) model, ZJU and SHH were more sensitive to emissions from eastern China (especially in Zhejiang, Anhui, Jiangsu, Jiangxi, and Fujian provinces) than GSN (see Fig.1 and Supplementary Fig. 2)*”.

c) Line 140: text would read better as “are lower than the total emission for all of China”. On line 139, perhaps “emissions” would be “emission”.

Response:

We thank the reviewer for the suggestion. As suggested by Reviewer #2, these sentences have been removed from our revised manuscript.

d) Lines 145 & 147: in my opinion, the paper is easier to digest if “CCl₄” were not used as an adjective, so often. Here, I suggest “consumption of CCl₄”

Response:

We thank the reviewer for the suggestion. The sentence in Lines 145 & 147 was revised as below, and other similar phrases were revised throughout the manuscript.

Lines 133–136 in the revised manuscript: “*Moreover, emissions of CCl₄ in eastern China ($7.6 \pm 3.1 \text{ Gg yr}^{-1}$) during 2021–2022 were significantly higher than the consumption of CCl₄ for dispersive uses (0.12 Gg yr^{-1} in 2021) as reported by China to the United Nations Environment Programme (UNEP, <https://wesr.unep.org>)*”.

Lines 136–137 in the revised manuscript: “*The consumption of CCl₄ for dispersive uses in China was $0.20\text{--}0.26 \text{ Gg yr}^{-1}$...*”.

e) Line 157: perhaps “air masses”

Response:

We thank the reviewer for the suggestion. The original sentence was revised accordingly as “*Additionally, air masses corresponding to high CCl₄ concentrations typically pass through these regions*” (**Lines 146–147**).

f) Line 159: here, “high emissions of CCl₄” reads much better in my opinion

Response:

Thank you for the suggestion. The sentence was revised accordingly as “*The high emissions of CCl₄ from Shandong, Zhejiang, and Jiangsu were generally consistent with...*” (Line 148)

g) Line 161: how about “indisputable emissions of CCl₄” (see <https://forum.wordreference.com/threads/indisputable-and-undisputed.1474726> for some thoughts in undisputable versus indisputable)

Response:

We thank the reviewer for the suggestion. The sentence was revised accordingly as “*our study also revealed indisputable emissions of CCl₄ from Jiangxi (1.1 ± 0.85 Gg yr⁻¹) and Fujian (0.99 ± 0.41 Gg yr⁻¹) provinces*” (Lines 150–151)

h) Line 169: perhaps a comma after “Wan et al.”

Response:

We thank you for the suggestion. We have corrected it in the text.

Lines 158–160 in the revised manuscript: “*The emissions of CCl₄ from emissive uses in China were estimated to be zero...and Wan et al.¹, as they set consumption of CCl₄ for dispersive uses to zero.*”

i) Line 177: suggest “associated emission gaps for CCl₄”.

Response:

The sentence was revised accordingly as “*Thus, further comprehensive bottom-up and top-down studies are strongly recommended to bridge the associated emission gaps for CCl₄*” (Lines 166–167).

j) Line 185: no need in my opinion to repeat the value of 6.6 +/- 2.7 given just above. This sentence is still long and a bit awkward even this is deletion. Perhaps the term CFC-11-eq Gg yr⁻¹ can be given once, early in the sentence, so that the reader could more easily focus on the numbers and various species. Finally, I suppose “(but for all of China)” applies only to HCFC-22; the sentence could be modified to that there is clarity about how this phrase should be applied.

Response:

We thank you for this pertinent suggestion. The repeat value of 6.6 +/- 2.7 has been deleted. However, as deleting “CFC-11-eq Gg yr⁻¹” for emissions of each ODS may make the sentence slightly difficult to interpret, we chose to retain “CFC-11-eq Gg yr⁻¹” at each instance.

Lines 175–181 in the revised manuscript: “*Moreover, ODP-weighted CCl₄ emissions in eastern China are several times higher than those of other major ODSs; for example, they are 17.6 times higher than the 0.38 ± 0.75 CFC-11-eq Gg yr⁻¹ of dichlorodifluoromethane (CFC-12) in 2019 in eastern China², 8.4 times the 0.79 (0.65–1.1) CFC-11-eq Gg yr⁻¹ of 1,1-dichloro-1-fluoroethane (HCFC-141b) in 2020*”

in eastern China³, and 1.4 times the 4.6 (3.9–5.3) CFC-11-eq Gg yr⁻¹ of chlorodifluoromethane (HCFC-22; but for all of China) in 2019⁴.

k) Line 206: suggest “potential sources of CCl₄”

Response:

Thank you for the suggestion. The sentence was revised accordingly as “A comprehensive measurement campaign comprising 456 exhaust samples from 17 industry sectors was performed to determine potential sources of CCl₄”. (Lines 195–196).

l) Line 207: suggest “This campaign” rather than “This study” to connect better to prior sentence

Response:

The sentence was revised accordingly as “This campaign shows that ...”. (Line 197)

m) line 225: suggest “high concentrations of CCl₄”.

Response:

We thank the reviewer for the suggestion. The sentence was revised to “The high concentrations of CCl₄ in these sectors may arise from...”. (Line 211)

n) lines 235 to 240: not sure what the connection of high concentrations of CCl₄ with low annual ravenous enterprises means. I wonder if either another sentence could be added explaining why the authors think this could be important, or else perhaps dropping this paragraph and the associated figures from Supplement. If the editor is reading this deep into the minor comments, and wants to shorten the paper, I think not only this paragraph but also lines 241 to 248 could be removed.

Response:

We agree with you about lines 235 to 240. We have dropped this paragraph. However, it is necessary to show the scale of the industrial sector observed in this study and its changing trends, which can help to assess the relative importance of these sectors’ contribution to CCl₄ emissions. Hence, lines 241 to 248 have been retained.

o) Lines 249 to 252: Sorry, I think this is a rather weak concluding paragraph. Here, I would add in the years of the observations, as well as words in plain language (not abbreviations) of the most egregious industrial sources of CCl₄.

Response:

Thank you for this insightful suggestion. The original Lines 249 to 252 were rewritten.

Lines 230–235 in the revised manuscript: “This study quantified substantial emissions of CCl₄ in eastern China (7.6 ± 3.1 Gg yr⁻¹) during 2021–2022 based on observations at two sites located in Zhejiang Province, China. By conducting

extensive sampling, we detected high concentrations of CCl₄ in the exhaust gases from manufacture of general purpose machinery and manufacture of raw chemical materials, and chemical products, implying that these previously unreported industrial sectors may be potential sources of CCl₄.”

q) The station location on Figure 1 was a bit hard to see based on a printed copy of the paper. I could only understand by enlarging the PDF. Also, the caption uses “cross” that I interpret to be “+”, rather than “X”.

Response:

Thanks for the suggestion. The symbol representing the location of the sampling point has been enlarged, and the original symbol, “X” has been replaced with “+”.

Fig. 1 Average emission sensitivity derived from FLEXPART simulations and the average emission sensitivity of ZJU and SHH stations compared to that of the GSN station in 2021 and 2022. a, b Average emission sensitivities from ZJU and SHH stations. c, d Average emission sensitivities from GSN station. e, f Ratio of the average emission sensitivity of ZJU and SHH stations to that of the GSN station. ZJU, SHH, and GSN sites are represented by the blue triangle, dot, and cross, respectively. The area framed by the bold black line is the target area for the inversion. The name of each Chinese province is given in Supplementary Fig. 1. Distribution of the emission sensitivities for a larger domain than that in Fig. 1 is shown in Supplementary Fig. 2.

In the interest of getting this back to the journal by the deadline, coupled with the holiday in my home country, I shall admit I did not review the Methods section in the revised paper. I did however read the responses to my prior comments carefully (I was reviewer 1), and I think the use of FLEXPART to compute the emission of CCL4 rather than the correlation method is a major step forward. I have no concerns about the methodology of the revised paper.

Response:

We are grateful to you for suggesting that we improve the emission quantification method used in this study in the first review round and for pointing out many minor errors in the second review round. Happy holidays.

END OF REVIEW

References:

- 1 Wan, D., Xu, J., Zhang, J., Tong, X. & Hu, J. Historical and projected emissions of major halocarbons in China. *Atmos. Environ.* **43**, 5822-5829 (2009).
- 2 Park, S. et al. A decline in emissions of CFC-11 and related chemicals from eastern China. *Nature* **590**, 433-437 (2021).
- 3 Western, L. M. et al. A renewed rise in global HCFC-141b emissions between 2017–2021. *Atmos. Chem. Phys.* **22**, 9601-9616 (2022).
- 4 Wu, J., Li, T., Wang, J., Zhang, D. & Peng, L. Establishment of HCFC-22 National-Provincial-Gridded Emission Inventories in China and the Analysis of Emission Reduction Potential. *Environ. Sci. Technol.* **56**, 814-822 (2022).

Responses to Reviewer #2:

It remains true that on a global scale substantial gaps still exist in our understanding of the measured global atmospheric decline of CCl₄ concentrations. Emissions derived from global atmospheric concentration changes (the top-down approach) are larger than can be explained by activity-based estimates of sources (bottom-up approach). Measurements of CCl₄ in regions dense with chemical production plants, especially in China, are highly valuable and have the potential to improve our understanding of these important questions regarding CCl₄ emissions. As a result, this work reporting high quality and high-frequency measurements at two new sites in China has the potential to add significantly to our understanding of these gaps and of the magnitude and nature of CCl₄ emissions coming from this important region of the world.

Response:

Thank you for recognizing the importance of this work. We have responded to each comment point-by-point.

The authors have made substantial improvements to their manuscript in response to reviewers' comments, especially in the methods used to derive emissions. In a few instances, however, they failed to address fundamental concerns expressed in the initial review (e.g., accurately conveying the goal of the "MEE supervision program", was it for better tracking of production and not, as the authors suggest, related to minimizing residual CCl₄ emissions). This point is central to how one might expect emissions from known processes to have evolved over time. As a result, I disagree with their central conclusion about the presence of a substantial gap in our understanding of CCl₄ emissions from China in the most recent years (2021-2022). In previous years, there clearly were substantial unexpected and unexplained CCl₄ emissions, and this point has been made in many previous studies. But the results presented here for 2021-2022, which give a better picture of total Chinese emissions than previous studies owing to the location of the new sites, suggest that there is no longer a gap in our understanding of Chinese emissions in this recent period. This is essentially the opposite message being conveyed in the current manuscript. Without substantial revision of the main points of this paper I cannot condone publication in any Journal.

Response:

Thank you for this comment. We agree that the total emissions of CCl₄ in China may not have decreased since the feedstock uses of CCl₄ may have substantially increased. As suggested, we have made "substantial revision of the main points in this paper".

As for the "MEE supervision program", CCl₄ emissions may decline. One factory was illegally producing CFC-11, and substantial discharges of CCl₄ (used as feedstock to produce CFC-11) were observed

(https://sd.ifeng.com/zbc/detail_2013_08/02/1064095_0.shtml; in Chinese). This

implies that CCl₄ emissions may decline after the MEE supervision program of controlling the uses of CCl₄ to produce CFC-11. However, we agree with your suggestion that CCl₄ emissions from non-controlled sources may increase in recent years, given the use of CCl₄ as feedstock to produce HFCs in larger amounts. Therefore, given the combined effects of the above activities on CCl₄ emissions, the latest changes in CCl₄ emissions in eastern China after 2018 remain unclear.

As for the statement that reads: “there is no longer a gap in our understanding of Chinese emissions in this recent period”, there are still gaps between top-down and bottom-up estimates of CCl₄ emissions in China. The CCl₄ emission in China in 2014 bottom-up estimated by Sherry et al.¹ was mainly from chloromethane production plants (6.6 Gg out of 7.3 Gg of total emissions in 2014), However, based on the field measurement-based emission factors, Li et al.² reported that CCl₄ emissions from chloromethane production plants were only 2.2 ± 1.6 Gg/yr in 2019, indicating that the emissions from chloromethane production plants and total emissions in China may have been overestimated by Sherry et al.¹ The total emissions from chloromethane production plants (2.2 ± 1.6 Gg/yr in 2019) and dispersive uses (0.21 Gg in 2021, <https://wesr.unep.org>) in China are smaller than our top-down emission estimates for eastern China. Thus, this study aimed to search the potential industrial source of CCl₄ emissions. Related revisions in the manuscript are given below:

Lines 74–84 in the revised manuscript: “*The “Supervision program” was implemented to improve the tracking of produced CCl₄, thereby preventing its conversion to CFCs (e.g., CFC-11) or its sale for controlled or emissive applications, which may lead to a decline in CCl₄ emissions from this source. CCl₄ is also used as feedstock, e.g., to produce pentafluoropropane (HFC-245fa) and fluoroolefins (HFOs) (these processes are not controlled by the Montreal Protocol)³. Since the feedstock use of CCl₄ for the production of HFCs, HFOs, and other chemicals has increased by ~70% between 2015 and 2019 in China³, an increase in CCl₄ emissions from feedstock uses is expected. Given the combined effects of the above activities, the latest emissions of CCl₄ in eastern China are unclear. However, no data....”*

Lines 187–193 in the revised manuscript: “*Based on the field measurement-based emission factors, Li et al.² reported that CCl₄ emission from chloromethane production plants was only 2.2 ± 1.6 Gg yr⁻¹ in 2019 in China. The consumption of CCl₄ for dispersive uses as reported by China to the UNEP (<https://wesr.unep.org>) was 0.12 Gg yr⁻¹ in 2021. Thus, the total emissions of CCl₄ from chloromethane production plants and dispersive uses in China are smaller than our top-down emission estimates, indicating the presence of other sources of CCl₄ emissions.”*

Based on this comment, the title of this article was also revised to “*Substantial CCl₄ emissions in eastern China during 2021–2022 and exploration of potential new sources*”.

Bottom-up analyses encompassing all known sources not controlled by the Montreal Protocol and, in my estimation, not likely to decrease over time are indicated as contributing 7.3 Gg/yr of emission in 2014 (lines 56-61). Because the MEE

supervision of CCl₄ production that was adopted after 2018 was focused primarily on tracking CCl₄ production in order to avoid any siphoning of this production to create CFCs, it seems unlikely to expect that it had any effect on residual, leakage-related emissions from non-controlled sources. The authors have implied that the MEE program should have caused emission reductions, but they failed to provide any evidence to back up this suggestion in their response. Hence, I have no reason to expect substantial declines in CCl₄ emissions in the years following 2014 from uncontrolled processes; in fact, I expect that they could have increased in recent years, given the use of CCl₄ as feedstock to produce HFCs in larger amounts in recent years.

Response:

We express our thanks for this suggestion. As for the impact of MEE supervision program on CCl₄ emissions, please see our response above. Meanwhile, we agree that CCl₄ emissions from the non-controlled sources may increase in recent years, given the use of CCl₄ as feedstock to produce HFCs in larger amounts. Therefore, given the combined effects of the above activities, the latest changes in CCl₄ emissions in eastern China, especially after 2018, remain unclear. Related description on “MEE supervision program” has been substantially revised accordingly.

Lines 71–84 in the revised manuscript: *“To strengthen the regulation of CFC-11 and production of CCl₄, the Ministry of Ecology and Environment of China has dispatched working groups to perform on-site supervision and installed online monitoring systems for enterprises producing CCl₄ as a by-product in the country⁴. The “Supervision program” was implemented to improve the tracking of produced CCl₄, thereby preventing its conversion to CFCs (e.g., CFC-11) or its sale for controlled or emissive applications, which may lead to a decline in CCl₄ emissions from this source. CCl₄ is also used as feedstock, e.g., to produce pentafluoropropane (HFC-245fa) and fluoroolefins (HFOs) (these processes are not controlled by the Montreal Protocol)³. Since the feedstock use of CCl₄ for the production of HFCs, HFOs, and other chemicals has increased by ~70% between 2015 and 2019 in China³, an increase in CCl₄ emissions from feedstock uses is expected. Given the combined effects of the above activities, the latest emissions of CCl₄ in eastern China are unclear....”*

The new results presented in this manuscript, which the authors rightly suggest provide a more complete picture of total emissions from China than studies relying on data only from Gosan (see Figure 1), yield a CCl₄ emission of 7.0-8.2 Gg/yr in 2021 and 2022, with uncertainties of about 3 Gg on each value. Given the consistency of these emissions magnitudes with expectations from uncontrolled sources (7.3 in 2014, and perhaps increasing after), it seems reasonable to conclude that there is no longer a gap in our understanding of CCl₄ emissions from China.

Response:

Thank you for this comment. We have provided our responses above; however, we have presented the responses here as well.

The CCl₄ emissions estimated in China in 2014 using the bottom-up approach by

Sherry et al.¹ were mainly sourced from chloromethane production plants (6.6 Gg out of 7.3 Gg of total emissions in 2014). However, the study by Li et al.² on field measurement-based emission factors stated that CCl₄ emissions from chloromethane production plants were only 2.2 ± 1.6 Gg/yr in 2019 in China, indicating the emissions from chloromethane production plants and total emissions in China may have been overestimated by Sherry et al.¹. The total emissions from chloromethane production plants (2.2 ± 1.6 Gg/yr in 2019) and dispersive uses (0.21 Gg in 2021, <https://wesr.unep.org>) in China are smaller than our top-down emission estimates. Therefore, there are still gaps between top-down and bottom-up estimates of CCl₄ emissions in China.

The additional findings discussed in this manuscript related to previously unidentified sources of CCl₄ is interesting and needs further study, but the evidence presented here suggests that those unaccounted sources are likely small relative to Chinese emission totals.

Response:

Thank you for this comment. Right now, we cannot reach the conclusion that those unaccounted sources are likely small relative to Chinese emission totals. Owing to lack of activity data and CCl₄ concentrations of the investigated industrial sectors for all of China, we are not in a position to determine the contribution of the investigated industrial sectors to total emissions of CCl₄ in China.

The number of enterprises has exhibited an obvious upward trend for the MGPM and MEMA industries since 2018 (growth rate: 10–16% yr⁻¹ during 2018–2021)⁵ (Supplementary Fig. 7). Although the number of enterprises in most other industrial sectors was relatively stable during 2011–2021⁶ (Supplementary Fig. 7), the overall trend has been on the rise (Supplementary Fig. 7). Considering that they may represent potential sources of CCl₄ emissions, attention should be paid to these industries (particularly MGPM) to evaluate their impact on the ozone layer.

We agree with your point that their contribution should be studied in the future.

In some sections of the manuscript, the authors suggest that the new measurements provide only estimates for eastern China, and that some earlier work (Park et al., 2018 and Lunt et al., 2018) argues for total Chinese emissions being substantially larger. But I do not think this is an accurate interpretation. The earlier studies cited by the authors are based on very uncertain extrapolations of measurements from the Gosan station, which the authors have clearly demonstrated to be (Figure 1) much less likely to represent emission magnitudes from China more broadly than the results from their new stations.

Response:

We agree with the comment. Accordingly, the original description “*Additionally, the CCl₄ emissions in eastern China reported in this study (7.6 ± 3.1 Gg yr⁻¹ in 2021–2022) and by Park et al.⁷ are lower than those for total China reported by Park et al.⁸ (23.6 ± 7.1 Gg yr⁻¹ during 2011–2015) and Lunt et al.⁹ (17 (11–24) Gg yr⁻¹ during*

2009–2016) (*Error! Reference source not found.b*), indicating the presence of CCl₄ emissions in non-eastern regions of China” has been removed.

In summary, this paper provides important new measurements data and better estimates of total Chinese CCl₄ emissions, and the new results are likely more accurately interpreted given the available information to suggest that our understanding of CCl₄ emissions from China have improved over time, not the opposite. What would strengthen the manuscript notably is a very close look at the bottom-up emission estimates from non-controlled processes to understand how they have evolved over time instead of presuming without careful consideration what influence the "supervision program" might have had on emission expectations for CCl₄ in years since 2018.

Response:

We thank you for the positive comments and feedback.

We do agree that there is no longer a gap in our understanding of CCl₄ emissions from China based on top-down methods, but the gap between top-down and bottom-up emission estimates is still large (see our responses above). Based on this review comment, we agree that the changes in CTC emissions cannot be judged by the execution of “supervision program”. CCl₄ is also used as feedstock, e.g., to produce pentafluoropropane (HFC-245fa) and fluoroolefins (HFOs), the processes for which are not controlled by the Montreal Protocol³. Since the feedstock use of CCl₄ for HFCs, HFOs and other chemicals has increased by ~70% between 2015 and 2019 in China, an increase in CCl₄ emissions from feedstock uses is expected³. Given the combined effects of the above activities, the latest emissions of CCl₄ in eastern China are unclear.

Lines 187–193 in the revised manuscript: “Based on the field measurement-based emission factors, Li et al.² reported that CCl₄ emission from chloromethane production plants was only 2.2 ± 1.6 Gg yr⁻¹ in 2019 in China. The consumption of CCl₄ for dispersive uses as reported by China to the UNEP (<https://wesr.unep.org>) was 0.12 Gg yr⁻¹ in 2021. Thus, the total emissions from chloromethane production plants and dispersive uses in China are smaller than our top-down emission estimates, indicating the presence of other sources of CCl₄ emissions.”

Lines 77–79 in the revised manuscript: “CCl₄ is also used as feedstock, e.g., to produce pentafluoropropane (HFC-245fa) and fluoroolefins (HFOs) (these processes are not controlled by the Montreal Protocol)³. Since the feedstock use of CCl₄ for the production of HFCs, HFOs, and other chemicals has increased by ~70% between 2015 and 2019 in China³, an increase in CCl₄ emissions from feedstock uses is expected.”

References:

- 1 Sherry, D., McCulloch, A., Liang, Q., Reimann, S. & Newman, P. A. Current sources of carbon tetrachloride (CCl₄) in our atmosphere. *Environ. Res. Lett.* **13**, 024004 (2018).

- 2 Li, B. et al. Emission factors of ozone-depleting chloromethanes during production processes based on field measurements surrounding a typical chloromethane plant in China. *J. Clean. Prod.* **414** (2023).
- 3 UNEP. Updated report on the production of CTC and its feedstock uses in China (decision 84/41(b) and (c)). Report No. UNEP/OzL.Pro/ExCom/90/9/Add.1, (United Nations Environment Programme, Montreal, 2022).
- 4 Ministry of Ecology and Environment of China. *Record of the regular press conference of the Ministry of Ecology and Environment in August 2019 (in Chinese)*, https://www.mee.gov.cn/xgk/2018/xgk/xxgk15/201908/t20190830_730891.html (2019).
- 5 National Bureau of Statistics (NBS). *China Statistical Yearbook*. (China Statistics Press, 2021).
- 6 National Bureau of Statistics (NBS). *China Statistical Yearbook*. (China Statistics Press, 2011-2021).
- 7 Park, S. et al. A decline in emissions of CFC-11 and related chemicals from eastern China. *Nature* **590**, 433-437 (2021).
- 8 Park, S. et al. Toward resolving the budget discrepancy of ozone-depleting carbon tetrachloride (CCl₄): an analysis of top-down emissions from China. *Atmos. Chem. Phys.* **18**, 11729-11738 (2018).
- 9 Lunt, M. F. et al. Continued Emissions of the Ozone-Depleting Substance Carbon Tetrachloride From Eastern Asia. *Geophys. Res. Lett.* **45**, 11423-11430 (2018).